# Attenuated PDGF signaling drives alveolar and microvascular defects in neonatal chronic lung disease

Prajakta Oak[1,†], Tina Pritzke[1,†], Isabella Thiel[1], Markus Koschlig[1], Daphne S Mous[2], Anita Windhorst[3], Noopur Jain[1,4], Oliver Eickelberg[1], Kai Foerster[5], Andreas Schulze[5], Wolfgang Goepel[6], Tobias Reicherzer[5], Harald Ehrhardt[7], Robbert J Rottier[2], Peter Ahnert[8], Ludwig Gortner[9], Tushar J Desai[10,*] [iD] & Anne Hilgendorff[1,4,5,11,**] [iD]

## Abstract

Neonatal chronic lung disease (nCLD) affects a significant number of neonates receiving mechanical ventilation with oxygen-rich gas (MV-O$_2$). Regardless, the primary molecular driver of the disease remains elusive. We discover significant enrichment for SNPs in the PDGF-Rα gene in preterms with nCLD and directly test the effect of PDGF-Rα haploinsufficiency on the development of nCLD using a preclinical mouse model of MV-O$_2$. In the context of MV-O$_2$, attenuated PDGF signaling independently contributes to defective septation and endothelial cell apoptosis stemming from a PDGF-Rα-dependent reduction in lung VEGF-A. TGF-β contributes to the PDGF-Rα-dependent decrease in myofibroblast function. Remarkably, endotracheal treatment with exogenous PDGF-A rescues both the lung defects in haploinsufficient mice undergoing MV-O$_2$. Overall, our results establish attenuated PDGF signaling as an important driver of nCLD pathology with provision of PDGF-A as a protective strategy for newborns undergoing MV-O$_2$.

**Keywords** bronchopulmonary dysplasia; neonatal chronic lung disease; PDGF-Rα; transforming growth factor-β; VEGF-A
**Subject Categories** Cardiovascular System; Respiratory System

## Introduction

Positive pressure mechanical ventilation with O$_2$-rich gas (MV-O$_2$) offers life-saving treatment for respiratory failure due to lung immaturity and insufficient respiratory drive in preterm infants. Unfortunately, this therapy significantly increases the risk for an important number of preterm infants and a subset of newborns to develop neonatal chronic lung disease (nCLD), that is, bronchopulmonary dysplasia (BPD; Hargitai et al, 2001; Merritt & Boynton, 2009; Jobe, 2011; Konig & Guy, 2014). Characterized by defective alveolar septation and impaired vascularization, nCLD is associated with poor pulmonary and neurological long-term outcomes in affected infants (Ehrenkranz et al, 2005; Doyle & Anderson, 2009). The adverse effects of positive pressure MV-O$_2$ on pulmonary development have been reproduced in experimental models of the disease (Scherle, 1970; Hamilton et al, 2003; Ehrenkranz et al, 2005; Bland et al, 2008; Hilgendorff et al, 2011), manifesting as increased lung apoptosis and disordered matrix elastin characteristic of nCLD. Because a myriad of developmental signals are perturbed in the setting of active disease, it is difficult to distinguish primary drivers of pathology from secondary effectors or compensatory responses. Yet, if a defect in a specific signaling pathway underlies the susceptibility to developing nCLD and orchestrates the multiple processes that execute disease pathology, it would be critical to identify this potential therapeutic target.

An essential role for platelet-derived growth factor (PDGF) signaling in alveolar development was established by the discovery that

---

1  Comprehensive Pneumology Center, University Hospital of the University of Munich and Helmholtz Zentrum Muenchen, Munich, Germany
2  Department of Pediatric Surgery, Erasmus Medical Center – Sophia Children's Hospital, Rotterdam, The Netherlands
3  Institute for Medical Informatics, Justus-Liebig-University, Giessen, Germany
4  Department of Pediatrics, Stanford University School of Medicine, Stanford, CA, USA
5  Department of Neonatology, Perinatal Center Grosshadern, Ludwig-Maximilians University, Munich, Germany
6  Department of General Pediatrics, University Clinic of Schleswig-Holstein, Campus Lübeck, Lübeck, Germany
7  Department of General Pediatrics and Neonatology, Justus-Liebig-University and Universities of Giessen and Marburg Lung Center (UGMLC), Giessen, Germany
8  Institute for Medical Informatics, Statistics, and Epidemiology (IMISE), University of Leipzig, Leipzig, Germany
9  Department of Pediatrics and Neonatology, Medical University Vienna, Vienna, Austria
10 Department of Internal Medicine, Pulmonary and Critical Care, Stanford University School of Medicine, Stanford, CA, USA
11 Center for Comprehensive Developmental Care, Dr. von Haunersches Children's Hospital, University Hospital Ludwig-Maximilians University, Munich, Germany
   *Corresponding author. Tel: +1 650 723 1696; Fax: +1 650 498 6288; E-mail: tdesai@stanford.edu
   **Corresponding author. Tel: +49 89 3187 4675; Fax: +49 89 3187 4661; E-mail: a.hilgendorff@med.uni-muenchen.de
   †These authors contributed equally to this work

---

lungs of "knockout" mice lacking PDGF-A failed to form alveoli, with animals that survived infancy demonstrating an emphysema-like phenotype of enlarged distal air sacs (Bostrom *et al*, 1996; Lindahl *et al*, 1997). This lung pathology was attributed to failure of migration of PDGF-Rα-positive alveolar smooth muscle progenitor cells (also known as myofibroblasts) into the distal embryonic lung. Because myofibroblasts are believed to drive normal subdivision of primitive air sacs into mature alveoli ("secondary septation"), their absence presumably resulted in abnormally large distal air sacs due to a failure to execute this process. Deletion of the cognate receptor, PDGF-Rα, resulted in death in mid-gestation before air sac morphogenesis initiated (Bostrom & Betsholtz, 2002), but transgenic rescue of the profound craniofacial abnormalities and spina bifida enabled survival through birth. Distal lungs in these mutants also lacked myofibroblasts and failed to undergo secondary septation (Sun *et al*, 2000).

When the PDGF signaling pathway was examined in the lungs of animal models of nCLD employing mechanical ventilation (MV-O₂), reduced abundance of PDGF-A and PDGF-Rα proteins or mRNA was observed (Bland *et al*, 2003, 2007, 2008), similar to the lungs of neonatal rats exposed to hyperoxia (Powell *et al*, 1992) and preterm infants developing nCLD (Popova *et al*, 2014). These findings suggest that reduced PDGF signaling may be involved in the air sac morphogenesis defect of nCLD, but a causal role has not yet been demonstrated. Furthermore, since microvascular defects were not reported in the lungs of PDGF mouse mutants that failed to undergo secondary septation, the position of this pathway in the hierarchy of perturbed signaling in nCLD is uncertain. Here, in a case–control study of infants with nCLD, we discover significant enrichment for SNPs in the PDGF-Rα gene associated with reduced PDGF-R levels and diminished migration of lung fibroblasts suggesting that impairment of this pathway might be a primary driver of disease. We confirm this model using gene-targeted mice haploinsufficient for PDGF-Rα, showing an interaction with an established model of MV-O₂ that reproduces the multiple pathologies of nCLD, all of which are ameliorated by exogenous administration of PDGF-A protein during MV-O₂. We also dissect the molecular crosstalk that mediate nCLD pathology, showing that attenuated PDGF-Rα signaling results in a reduction of VEGF-A expression, likely responsible for the vasculature phenotype through increased endothelial cell apoptosis. We also find that increased TGF-β and mechanical stretch, driving lung injury in the newborn lung, act in concert to reduce PDGF-Rα signaling, exacerbating the underlying basal reduction in PDGF-Rα to a level sufficient to result in disease. Our work implicating attenuated PDGF signaling as a driver of the air sac and vasculature defects of nCLD, along with our demonstration that both pathologies can be ameliorated by exogenous PDGF-A protein, provides a strong rationale for pursuing augmentation of PDGF signaling as a potential targeted therapy for this serious disease.

# Results

## Enrichment of PDGF-Rα SNPs associated with reduced protein levels and migration of lung fibroblasts from ventilated preterm infants developing nCLD

We confirm reduced PDGF-Rα expression in lung fibroblasts isolated from ventilated preterm infants. This reduction in PDGF-Rα expression was correlated with increased duration of MV-O₂, as quantified by immunofluorescence and immunoblot analysis (Fig 1A). Further, a case–control analysis of PDGF-Rα in 1,061 newborns ($n = 492$ with moderate or severe BPD) identified 14 SNPs out of 117 with nominal significance ($P \leq 0.05$; Fig 1B). Three of these SNPs were highly statistically significant, with *P*-values below 0.001 (Appendix Table S1). The presence of at least one SNP at these three positions (rs12506783) was associated with reduced PDGF-Rα gene expression and PDGF-R as well as VEGF-A protein levels in blood from ventilated preterm infants (Fig 1C–E, Appendix Fig S1). In human lung fibroblast, this SNP is associated with reduced PDGF-Rα protein level in accordance with reduced migration (Fig 1F and G). Analysis of SNPs cis-regulating gene expression for genes in the PDGF pathway and its downstream pathways showed enrichment of low *P*-values for SNPs linked to the MAPKKK-cascade, JAK/STAT-cascade, apoptosis, cell cycle, DNA metabolism, lipid metabolism, protein metabolism, and actin and calcium ion homeostasis (Appendix Table S2). Together, these experiments indicate that MV-O₂ in humans results in a reduction of PDGF-Rα expression by alveolar fibroblasts, and suggests that genetic risk factors for reduced PDGF signaling in human infants is associated with an increased risk of developing nCLD.

## PDGF-Rα haploinsufficiency drives the air sac pathology of nCLD in neonatal mice undergoing MV-O₂

In order to test whether attenuated PDGF signaling was indeed a risk factor for the development of nCLD as suggested by our human SNP data and not restricted to the air sac component, we obtained gene-targeted mice lacking one allele of PDGF-Rα and subjected them to MV-O₂ using a unique preclinical mouse model. We found that lungs of PDGF-Rα haploinsufficient newborn mice undergoing MV-O₂ for 8 h showed a significant increase in distal airspace size and decrease in radial alveolar counts resembling nCLD pathology as assessed by quantitative morphometry when compared to unventilated controls (Fig 2A–C), whereas their ventilated WT littermates were unaffected. As expected with this phenotype, there were fewer secondary septae in PDGF-Rα$^{+/-}$ neonatal mice after MV-O₂ for 8 h as compared to WT mice (Fig 2D), with no significant difference in lung volumes between the groups (WT control 55.3 ± 7.0 μl/g bw; WT MV-O₂ 56.2 ± 14.2 μl/g bw; PDGF-Rα$^{+/-}$ control 59.7 ± 11.9 μl/g bw; PDGF-Rα$^{+/-}$ MV-O₂ 57.6 ± 23.7 μl/g bw; mean and SD each). Atelectasis, as analyzed using ImageJ, involved 14–18% of the total lung in both groups undergoing MV-O₂ (WT MV-O₂ 17.2 ± 9.6%; PDGF-Rα$^{+/-}$ MV-O₂ 22.8 ± 10.4%; $P = 0.61$).

Immunoblot and mRNA analysis confirmed reduced pulmonary PDGF-Rα level (Figs 2G and EV1D), reflected by the reduced number of myofibroblasts localized on septal crests in ventilated PDGF-Rα$^{+/-}$ mice lungs compared to WT littermates (Fig 2E and F). Diminished JAK-2 and STAT-3 in ventilated PDGF-Rα$^{+/-}$ mice reflects reduced PDGF-Rα downstream signaling (Fig 2H and I).

## PDGF-Rα haploinsufficiency drives reduced pulmonary micro-vessel density with increased endothelial cell apoptosis in neonatal mice undergoing MV-O₂

We next asked whether attenuated PDGF signaling would also reproduce the vascular defect of nCLD. Indeed, we demonstrated by

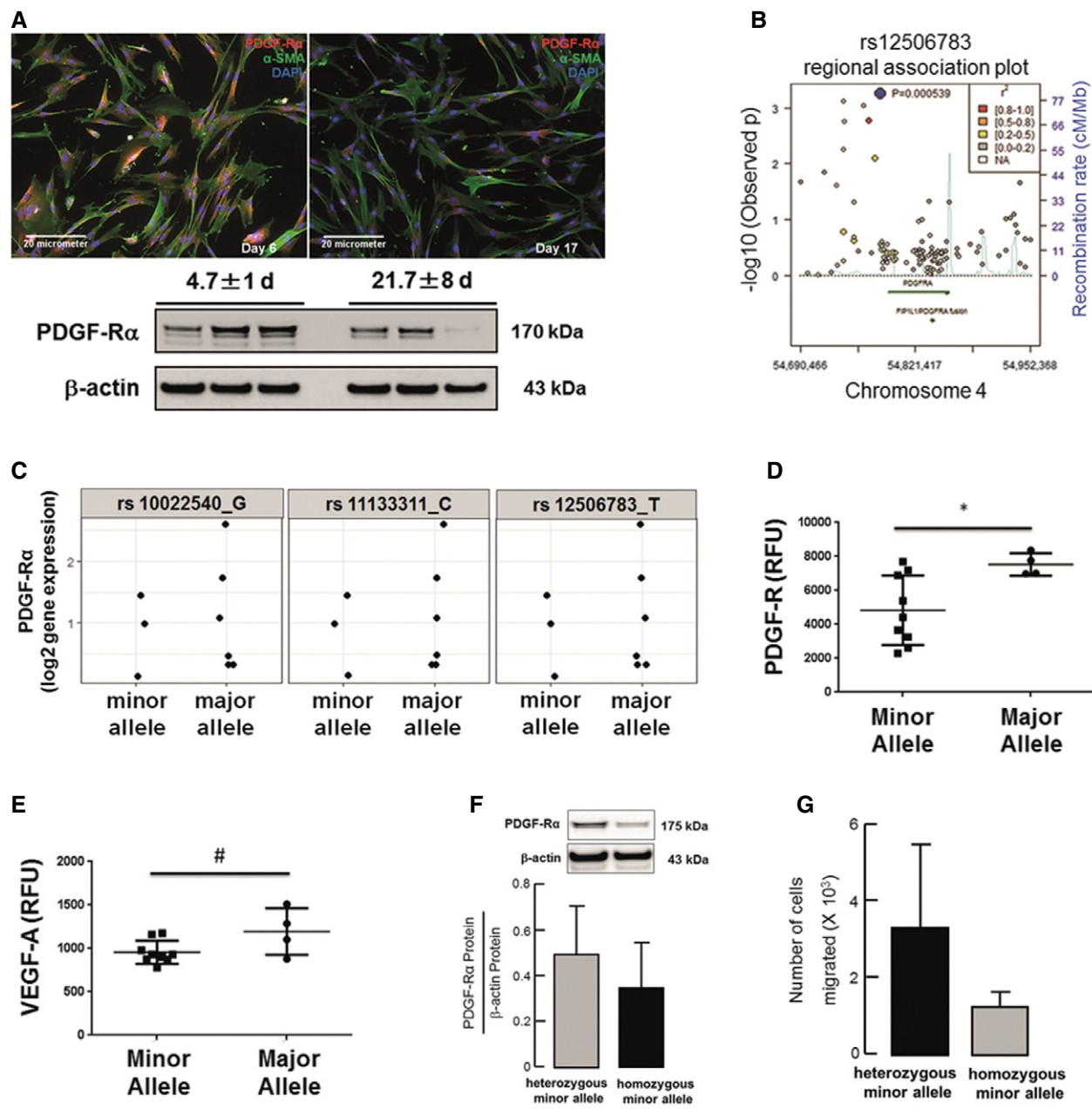

**Figure 1.  Enrichment of PDGF-Rα SNPs associated with reduced protein levels and migration of lung fibroblasts from ventilated preterm infants developing nCLD.**

A    Decreased PDGF-Rα expression in human lung fibroblasts (hMFBs) from preterms undergoing MV-O$_2$ (21.7 ± 8 vs. 4.7 ± 1 day of life; serial samples 2 & 5, 3 & 6; $n$ = 3 patients/group).

B    Regional association plot showing −log10 P-values ($y$-axis) of SNPs according to chromosomal positions ($x$-axis). Light blue: estimated recombination rate (cM/Mb, HapMap CEU population); blue: most significant SNP (rs12506783); red: $r^2 \geq 0.8$; orange: $0.8 > r^2 \geq 0.5$, yellow: $0.5 > r^2 \geq 0.2$, gray: $0.2 > r^2 \geq 0$. P-values were determined using R by case-control analysis with a logistic regression model including case/control status, sex, gestational age at birth, status "small for gestational age", and country of origin of the mother. Analysis was adjusted for relatedness to account for multiple births (R-package GenABEL).

C    Levels of PDGF-Rα gene expression in patients ($n$ = 9), which are carrying at least one SNP (minor allele) compared to patients with no SNPs (major allele). Major alleles are given in the figure labels. Minor alleles in rs10022540 are A, in rs11133311 are T, and in rs12506783 are C.

D, E    PDGF-R (D) and VEGF-A (E) protein levels in separate patient cohort ($n$ = 13) carrying at least one SNP (minor allele) at position rs12506783 compared to patients with no SNPs (major allele). Protein levels were quantified using SOMAlogic technique. Data are presented as mean ± SD. Two-tailed unpaired Student's $t$-test (*$P$ = 0.0336; #$P$ = 0.0863).

F, G    Representative PDGF-Rα levels (F) and migratory potential assessed by Boyden chamber assay (G) in fibroblasts isolated from tracheal aspirates of patients with nCLD. The fibroblast carrying SNP at both alleles (homozygote minor allele) displayed reduced PDGF-Rα levels and migration when compared to fibroblasts from patients carrying SNP at one allele (heterozygote minor allele). Data are presented as mean ± SD ($n$ = 3/4 replicates).

Source data are available online for this figure.

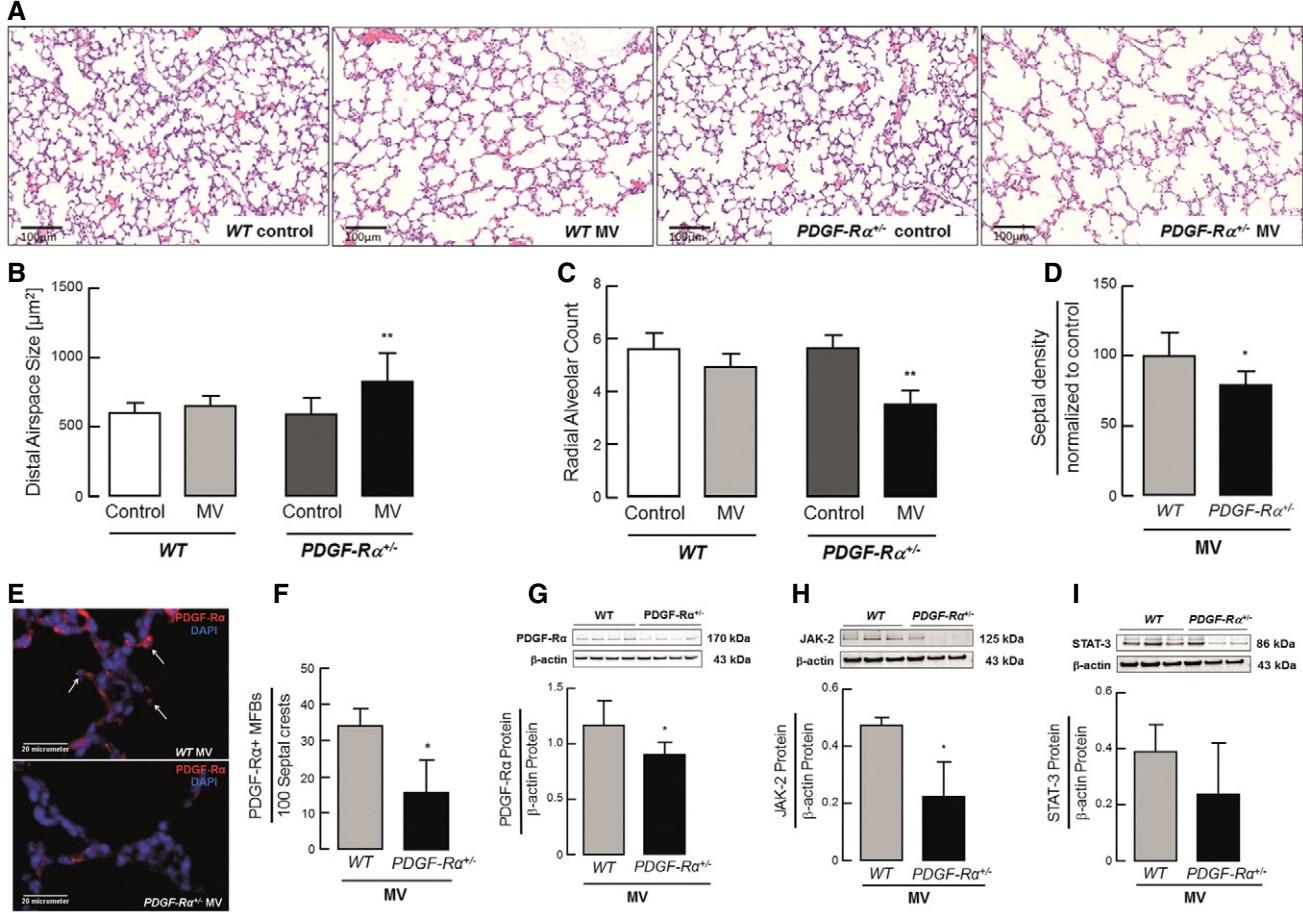

**Figure 2. PDGF-Rα haploinsufficiency drives the air sac pathology of nCLD in neonatal mice undergoing MV-O₂.**

A   Representative lung tissue sections (200×) from 5–8-day-old PDGF-Rα$^{+/+}$ (WT) and PDGF-Rα$^{+/-}$ mice after 8 h of MV-O₂ showing increased air space size compared to respective controls (O₂-control) spontaneously breathing 40% O₂ for 8 h.

B   Quantitative analysis of lung tissue sections showed increased alveolar area after 8 h of MV-O₂ in PDGF-Rα$^{+/-}$ mice, whereas no significant change was observed in WT mice when compared to respective controls ($n$ = 6–11 mice/group).

C   Radial alveolar counts (alveolar number) in lung tissue sections from WT and PDGF-Rα$^{+/-}$ mice were reduced after 8 h of MV-O₂ when compared to respective controls ($n$ = 6–11 mice/group).

D   Septal density was significantly reduced in PDGF-Rα$^{+/-}$ mice when compared to WT littermates after 8 h of MVO₂ ($n$ = 6–8 mice/group).

E   Immunofluorescence staining (400×, merged) for PDGF-Rα (red, white arrows; blue: DAPI) with decreased stain from the septal crests in lungs of ventilated PDGF-Rα$^{+/-}$ (lower panel) and WT (upper panel) mice undergoing MV-O₂.

F   Quantitative analysis of the immunofluorescence images showed reduced number of PDGF-Rα$^+$ myofibroblasts located at the septal crests (presented myofibroblasts number per 100 septal crests; 10 fields of view in PDGF-Rα and α-smooth muscle actin co-stained sections/animal, 4 animals/group).

G–I   Immunoblot analysis of PDGF-Rα (G) and its downstream proteins JAK-2 (H) and STAT-3 (I) showing a significant reduction in protein level in PDGF-Rα$^{+/-}$ neonatal mice in contrast to WT mice after MV-O₂ for 8 h ($n$ = 3 mice/group). PDGF-Rα levels are displayed as fold change of control. Panels (H) and (I) are from same blot hence having same β-actin bands.

Data information: In (B–D) and (F–I), data are presented as mean ± SD. **$P$ < 0.01, *$P$ < 0.05. Statistical analysis was performed for (B–D) by ordinary one-way ANOVA with Bonferroni's correction ($P$ = 0.0012–0.0293) and for (F–I) by two-tailed unpaired Student's $t$-test or Mann–Whitney test ($P$ = 0.024–0.028).

Source data are available online for this figure.

histological analysis of PDGF-Rα$^{+/-}$ mice undergoing MV-O₂, a significant reduction in the number of alveolar micro-vessels (20–100 μm), notably exceeding the effect observed in WT pups (Fig 3A). This finding was corroborated by immunoblot analysis showing reduced lung protein levels of the endothelial cell markers, VEGF-R2 (Fig 3B), and VE-cadherin (Fig 3C) in ventilated neonatal PDGF-Rα$^{+/-}$ mice, suggesting a net loss of endothelial cells. Co-staining for apoptosis and endothelial cell markers demonstrated significantly increased vascular endothelial cell death in the lung

periphery of ventilated neonatal PDGF-Rα$^{+/-}$ mice compared with WT littermates (Fig 3E and F).

To further explore the link between reduced PDGF signaling and endothelial cell apoptosis, we focused on VEGF-A, a critical regulator of pulmonary microvascular development (Nauck *et al*, 1997; Compernolle *et al*, 2002; Kamio *et al*, 2008; Ding *et al*, 2010; Prochilo *et al*, 2013). With MV-O₂, pulmonary VEGF-A protein level was significantly reduced in PDGF-Rα$^{+/-}$ mice compared to WT littermates (Fig 3D). Interestingly, vessel number

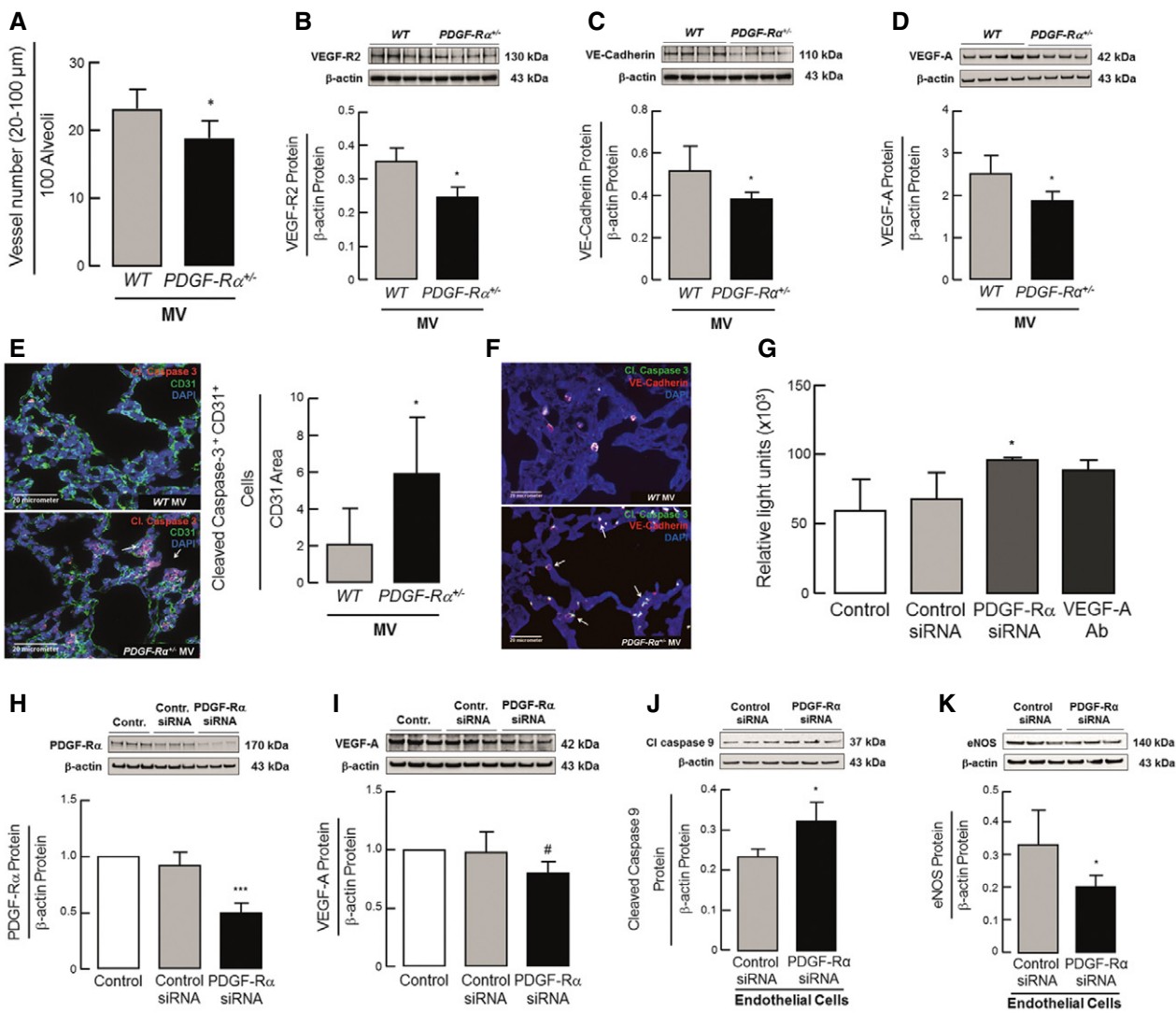

**Figure 3. PDGF-Rα haploinsufficiency drives reduced pulmonary micro-vessel density with increased endothelial cell apoptosis in neonatal mice undergoing MV-O₂.**

A–D  Histologic and immunoblot analysis displayed reduced small vessel number (20–100 μm diameter) normalized to 100 alveoli as well as reduced pulmonary VEGF-R2, VE-cadherin, and VEGF-A protein levels, respectively (n = 6–8 mice/group). Panels (B) and (C) are from same blot hence having same β-actin bands.

E  Immunofluorescence images of lung tissue (400×; merged) from neonatal PDGF-Rα$^{+/-}$ mice indicating increased cleaved caspase-3 (red, white arrows; lower panel) after 8 h of MV-O₂ in contrast to WT mice (upper panel; green: CD31; blue: DAPI). Double stain revealed increased cleaved caspase-3$^+$/CD31$^+$ cells normalized to CD31 area in PDGF-Rα$^{+/-}$ mice after 8 h of MV-O₂ (n = 4 mice/group, 4 sections/mice, and 10 images/section).

F  Representative image confirming increased endothelial apoptosis in neonatal PDGF-Rα$^{+/-}$ mice after 8 h of MV-O₂ (lower panel; white arrows) with VE-cadherin (red) and cleaved caspase-3 (green) and nucleus stained with DAPI (blue) when compared to WT mice (upper panel) (n = 2 mice/group).

G  Increased caspase-3 activation in HUVECs upon incubation with supernatants for 6 h obtained from lung mouse myofibroblasts after PDGF-Rα siRNA treatment when compared to control siRNA (n = 3 experiments).

H, I  *In vitro* application of PDGF-Rα siRNA to primary lung mouse myofibroblasts from WT mice diminished PDGF-Rα (H) protein (normalized to control), associated with reduced VEGF-A protein (I) (n = 3 mice/group).

J, K  Increased cleaved caspase-9 and reduced eNOS protein levels in HUVECs upon incubation with supernatants for 6 h obtained from lung mouse myofibroblasts after PDGF-Rα siRNA treatment when compared to control siRNA (n = 3 mice/group).

Data information: Data are presented as mean ± SD. ***$P < 0.001$, *$P < 0.05$ vs. control, $^\#P$ vs. control siRNA. Statistical test in (A–E) is two-tailed unpaired Student's *t*-test or Mann–Whitney test, in (J, K) one-tailed Mann–Whitney test ($P = 0.02$–$0.05$), in (G) is Kruskal–Wallis *H*-test ($P = 0.027$), and in (H, I) is ordinary one-way ANOVA with Bonferroni's correction ($^\#P = 0.0889$).

Source data are available online for this figure.

did not differ compared to unventilated PDGF-Rα$^{+/-}$ mice (Appendix Fig S2).

In order to determine whether this reduction was a direct consequence of reduced PDGF-Rα signaling in myofibroblasts, we isolated primary lung myofibroblasts from WT mice and measured their production of VEGF-A at baseline as well as following siRNA mediated knockdown of PDGF-Rα expression *in vitro* (Fig 3H and I). We also administered conditioned supernatant from PDGF-Rα

siRNA-treated mouse myofibroblasts to human umbilical vein endothelial cells (HUVECs) in culture, and found reduced endothelial cell survival due to an increase in apoptosis (Fig 3G and J) and a reduction in the permeability factor eNOS (Fig 3K), comparable to the effect seen with an anti-VEGF-A antibody. These experiments together with Fig 2G–I indicate that PDGF-Rα signaling in the myofibroblasts promotes VEGF-A expression, and corroborates that the microvascular phenotype is downstream of attenuated PDGF signaling.

### Supplemental PDGF-A rescues both the air sac and microvascular nCLD phenotypes induced by MV-O$_2$ in neonatal PDGF-Rα haploinsufficient mice

As suggested by the human SNP data and our experiments above, if the attenuated PDGF signaling drives the pathogenesis of nCLD in all its manifestations, experimentally augmenting PDGF signaling in the lungs of ventilated haploinsufficient mice should ameliorate both the air sac and microvascular phenotypes. To directly test this prediction, we administered exogenous PDGF-A protein by endotracheal delivery at the onset of MV-O$_2$. The results showed both an increase in peripheral lung micro-vessel number (20–100 μm) in treated vs. un-treated PDGF-Rα$^{+/-}$ neonatal mice undergoing MV-O$_2$, as well as a normalization of air sac defects, with increased alveolar and micro-vessel number compared with untreated controls (Fig 4A–C). The reduction in JAK-2, STAT-3, VE-cadherin, and VEGF-A protein expression in the lungs of ventilated mice was also ameliorated with PDGF-A treatment, supporting reversal of the endothelial cell apoptosis (Fig 4E–H). PDGF-A treatment further enhanced PDGF-Rα protein, associated with increased levels of AKT, suggesting a feed-forward mechanism where increased endosomal internalization leads to PDGF-Rα recycling in lung myofibroblasts with subsequent increase in receptor expression (Wang *et al*, 2004; Heldin, 2013) (Fig 4D and I). The lung periphery of mechanically ventilated PDGF-Rα$^{+/-}$ neonatal mice also exhibited an increase in (secreted) VEGF-A protein immunolocalized near myofibroblasts, providing further support (Fig 4J and M, upper panel). Quantification of immunofluorescent staining confirmed a pronounced reduction in apoptotic (cleaved caspase-3 positive) surface area and individual cells, with dramatically increased CD31 surface area in PDGF-A-treated mice

(Fig 4K–M lower panel). A physiological rescue was also observed, with an improvement in quasi-static compliance by lung function testing in neonatal mice pretreated with PDGF-A (Fig 4N), with no significant difference in lung volume (PDGF-A 49.8 ± 5.6 μl/g bw vs. no-PDGF-A 49.3 ± 3.9 μl/g bw; mean ± SD, *P* = 0.99, two-tailed Mann–Whitney test).

### Elevated TGF-β levels causally relate with reduced PDGF-Rα expression in nCLD patients and mice undergoing MV-O$_2$, reducing downstream signaling and migration in pulmonary myofibroblasts

Clinical and experimental studies have consistently demonstrated activation of pulmonary TGF-β signaling with MV-O$_2$ (Assoian *et al*, 1987; Groneck *et al*, 1994; Zhao *et al*, 1997; Yamamoto *et al*, 2002; Schultz *et al*, 2003; Vozzelli *et al*, 2004; Xu *et al*, 2006; Wu *et al*, 2008). We therefore investigated how reduced PDGF signaling and its demonstrated consequences are provoked in the neonatal lung. Hence, we characterized the two potential players mechanical stretch and TGF-β alone and in combination, with respect to their effect on primary mouse and human lung myofibroblasts.

Immunofluorescent images of tissue sections from nCLD patients (*n* = 7) showed reduced expression of PDGF-Rα associated with increased expression of pSMAD-2 compared with a control lung, supporting the relevance of the findings to patients suffering from nCLD (Fig 5A). In parallel, we performed gene expression microarray analysis of blood samples obtained from 20 preterm infants in the first 72 h after birth, which similarly demonstrated an inverse correlation between the levels of TGF-β1 and PDGF-Rα in patients who went on to develop nCLD that was not observed in patients who did not (Fig 5B). We next analyzed TGF-β signaling activity in the lungs of ventilated WT mice, which demonstrated increased pSMAD 2/3 protein with a concomitant reduction in PDGF-Rα$^+$ alveolar myofibroblasts (Fig 5C). To test the impact of TGF-β on PDGF-Rα promoter activity, we conducted a luciferase assay by transfecting CCL206 cells with a pGal vector carrying a PDGF-Rα promoter insert. Administration of TGF-β caused a 50% reduction in luciferase activity (Fig 5D), indicating that TGF-β affects PDGF-Rα gene transcription, supporting the inverse relationship observed in nCLD is causal.

---

**Figure 4. Supplemental PDGF-A rescues both the air sac and microvascular nCLD phenotypes induced by MV-O$_2$ in neonatal PDGF-Rα haploinsufficient mice.**

A–C  Improved alveolar structure in 5–8-day-old PDGF-Rα$^{+/-}$ mice undergoing 8 h of MV-O$_2$ after intra-tracheal treatment with PDGF-A (10 μl/g bw, 25 ng/ml PDGF-A) when compared to mice receiving sterile saline (200×), confirmed by quantitative image analysis with increased alveolar counts (B) as well as vessel number normalized to 100 alveoli (C) (20–100 μm; *n* = 2–4 mice/group).

D–I  Immunoblot analysis of total lung homogenates showed increased PDGF-Rα (D) together with increased JAK-2 (E), STAT-3 (F), VEGF-A (G), VE-cadherin (H), and AKT (I) protein levels in PDGF-A-treated PDGF-Rα$^{+/-}$ mice after 8 h of MV-O$_2$ when compared to WT littermates (*n* = 3–4 mice/group). Panels (D, H) and (E, F) are from same blot hence having same β-actin bands.

J–M  Quantitative image analysis indicated increased VEGF-A to PDGF-Rα protein levels (J, M upper panel) together with an increase in CD31 expression in relation to total tissue (K, M lower panel) and a decrease in apoptotic (cleaved caspase-3) CD31-expressing cells (L) in the lungs of PDGF-A-treated PDGF-Rα$^{+/-}$ mice when compared to saline-treated controls after 8 h of MV-O$_2$ (*n* = 2–4 mice/group). Upper panel in (M) shows sections from PDGF-Rα$^{+/-}$ mice treated with NaCl (left) or PDGF-A (right) stained with VEGF-A (green), PDGF-Rα (red) dual positive (orange, white arrows), and inserts show VEGF-A stain (green). Lower panel shows sections from PDGF-Rα$^{+/-}$ mice treated with NaCl (left) or PDGF-A (right) stained with cleaved caspase-3 (green), CD31 (red) dual positive (orange, white arrows). Nucleus is stained with DAPI (blue).

N  Treatment with PDGF-A in ventilated neonatal PDGF-Rα$^{+/-}$ mice led to improved lung compliance displayed as a function of airway pressure (Ptramax) and tidal volume when compared to untreated mice (*n* = 4 mice/group).

Data information: In (B–L, N), the data are presented as mean ± SD. ***P* < 0.001, ***P* < 0.01, *P* < 0.05, #*P* < 0.067. Statistical test used is one-tailed unpaired Student's *t*-test (*P* = 0.0001–0.065), in (C) and (G) is two-tailed Mann–Whitney and unpaired Student's *t*-test (*P* = 0.02–0.067).
Source data are available online for this figure.

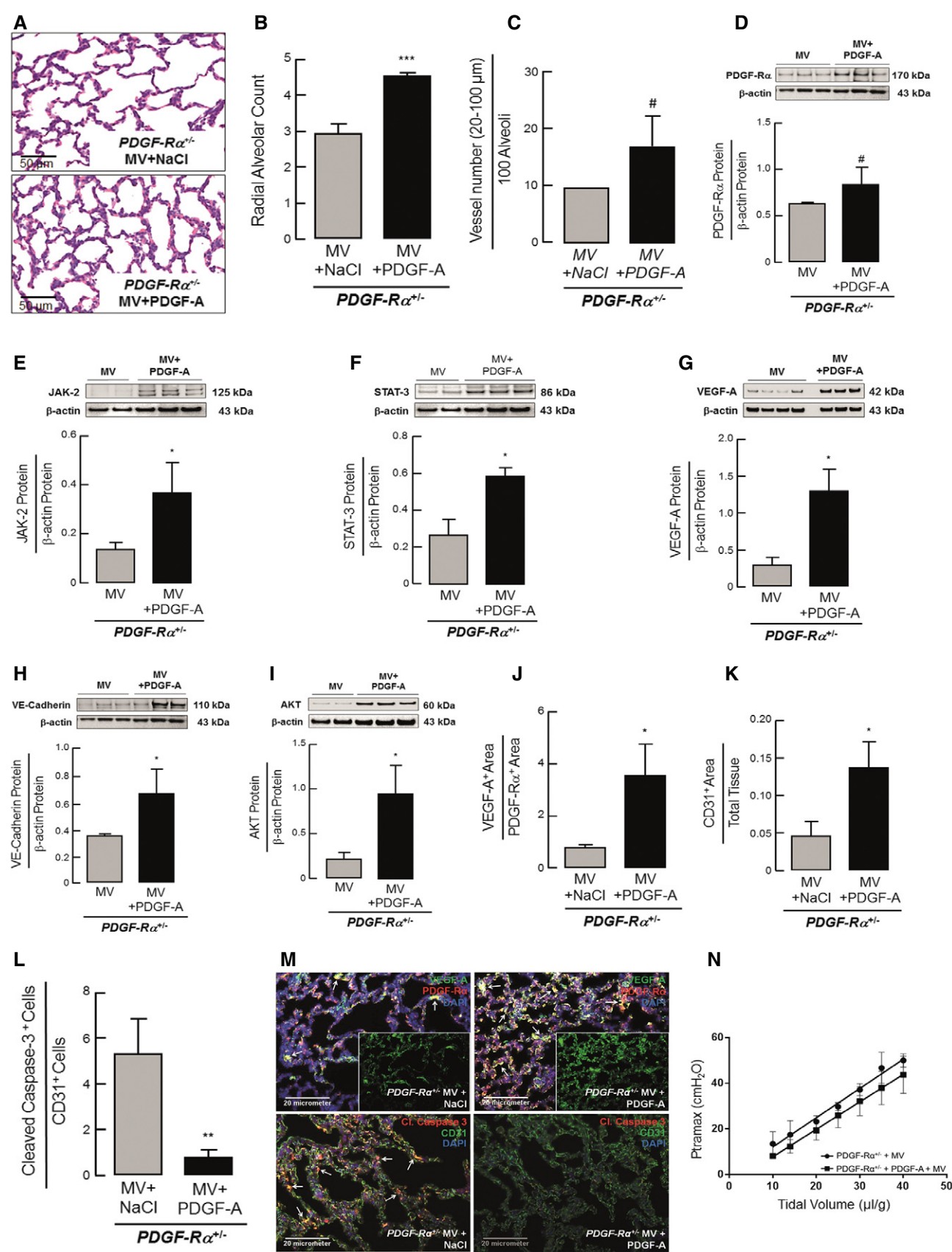

**Figure 4.**

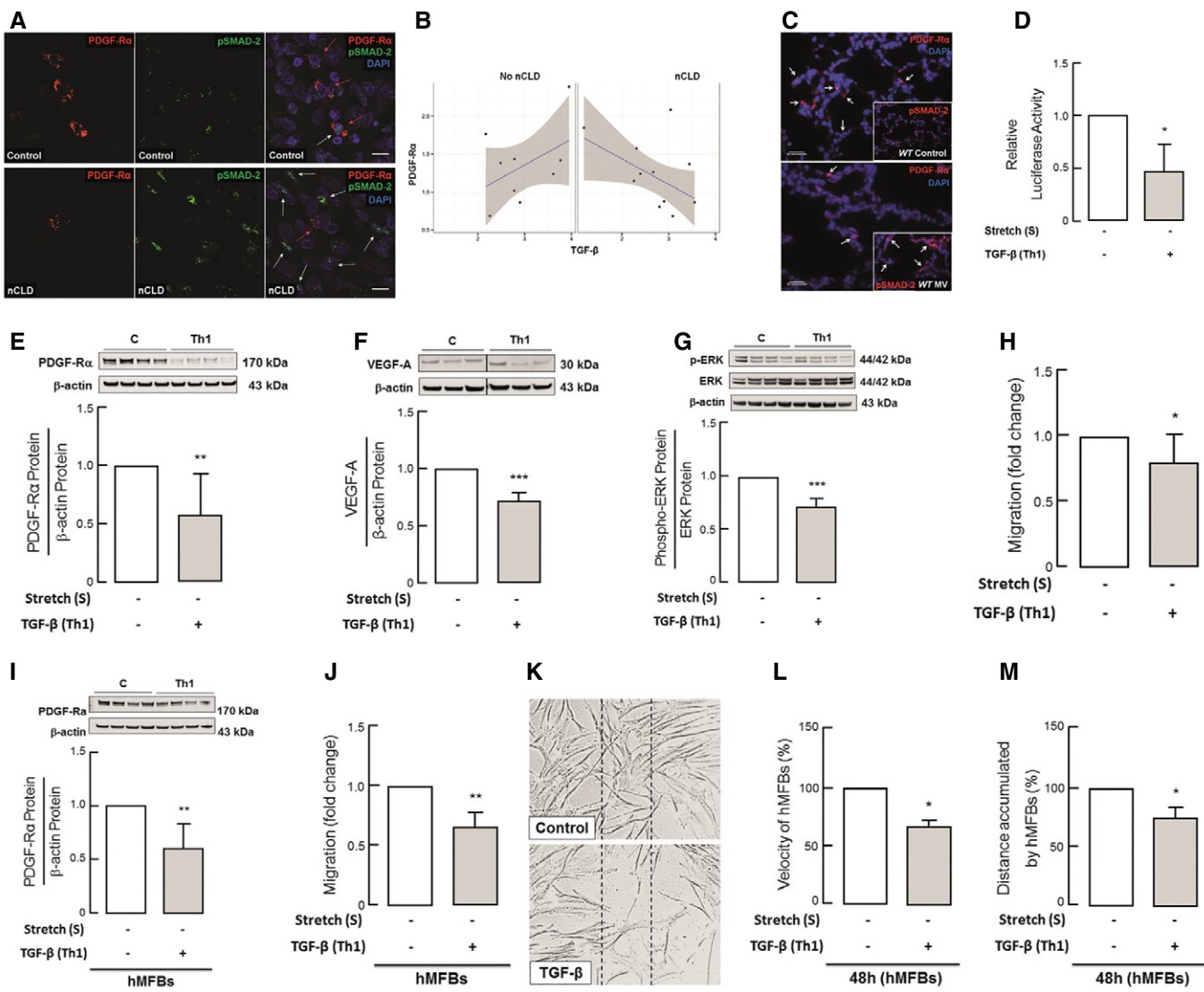

**Figure 5.  Elevated TGF-β levels causally relate with reduced PDGF-Rα expression in nCLD patients and mice undergoing MV-O₂, reducing downstream signaling and migration in pulmonary myofibroblasts.**

A       Representative immunofluorescence images showing reduced expression of PDGF-Rα (red) in the lungs of patients (*n* = 7) developing nCLD (lower left panel, red stain; white arrows) together with increased pSMAD-2 expression (lower middle panel, green stain; white arrows) as compared to lung sections from a non-nCLD patient (*n* = 1) (upper panel) (200×).

B       Negative correlation between PDGF-Rα and TGF-β1 in transcriptome analysis 72 h after birth in preterms with nCLD (*n* = 11) in contrast to non-nCLD (*n* = 9); *z* test of the difference of the Fisher's *z* transformed correlations divided by the standard error of the difference (*P* = 0.048); scatter plots log2-gene expression; linear regression (blue), with 95% CI (gray).

C       MV-O₂ reduced lung PDGF-Rα (main: red stain; white arrows) and increased pSMAD-2 levels (insert: red stain; white arrows) in ventilated neonatal WT (lower panel) when compared to control WT mice (upper panel); (*n* = 4 mice/group; 10 images/mouse; 200×).

D       Luciferase assay of CCL-206 cells transfected with pGL4.14 containing PDGF-Rα promoter revealing reduced promoter activity upon TGF-β application (normalized to control) (*n* = 3 experiments).

E–G    Immunoblot analysis showing reduced PDGF-Rα (E), VEGF-A (F), and pERK/EKR (G) protein levels upon TGF-β application alone in primary pulmonary myofibroblasts from 5–7-day-old WT mice (*n* = 6–9 mice/group).

H       Reduced migration of myofibroblasts (MFBs) from neonatal WT mice upon TGF-β application alone (*n* = 5 mice/group, 3 technical replicates).

I, J    Translation of the results in fibroblasts isolated from tracheal aspirates of ventilated preterm infants (hMFBs) displayed reduced PDGF-Rα levels (I) and migration assessed by Boyden chamber assay (J) upon TGF-β application (*n* = 3–5 patients/group).

K–M    Representative phase contrast images (100×) of scratch migration assays in human lung fibroblasts (hMFBs) after 48 h of TGF-β incubation indicating decreased wound closure (K) quantified by reduced velocity (L) and distance travelled (M) (*n* = 3 patients/group).

Data information: In (D–J) and (L, M), data are presented as mean ± SD and normalized to control. Statistical test used is two-tailed unpaired Student's *t*-test or Mann–Whitney test (*P* = 0.0002–0.039). \*\*\**P* < 0.001, \*\**P* < 0.01, \**P* < 0.05. C, un-stretched untreated control; Th1, un-stretched myofibroblasts subjected to 5 ng/ml TGF-β (24 h).

Source data are available online for this figure.

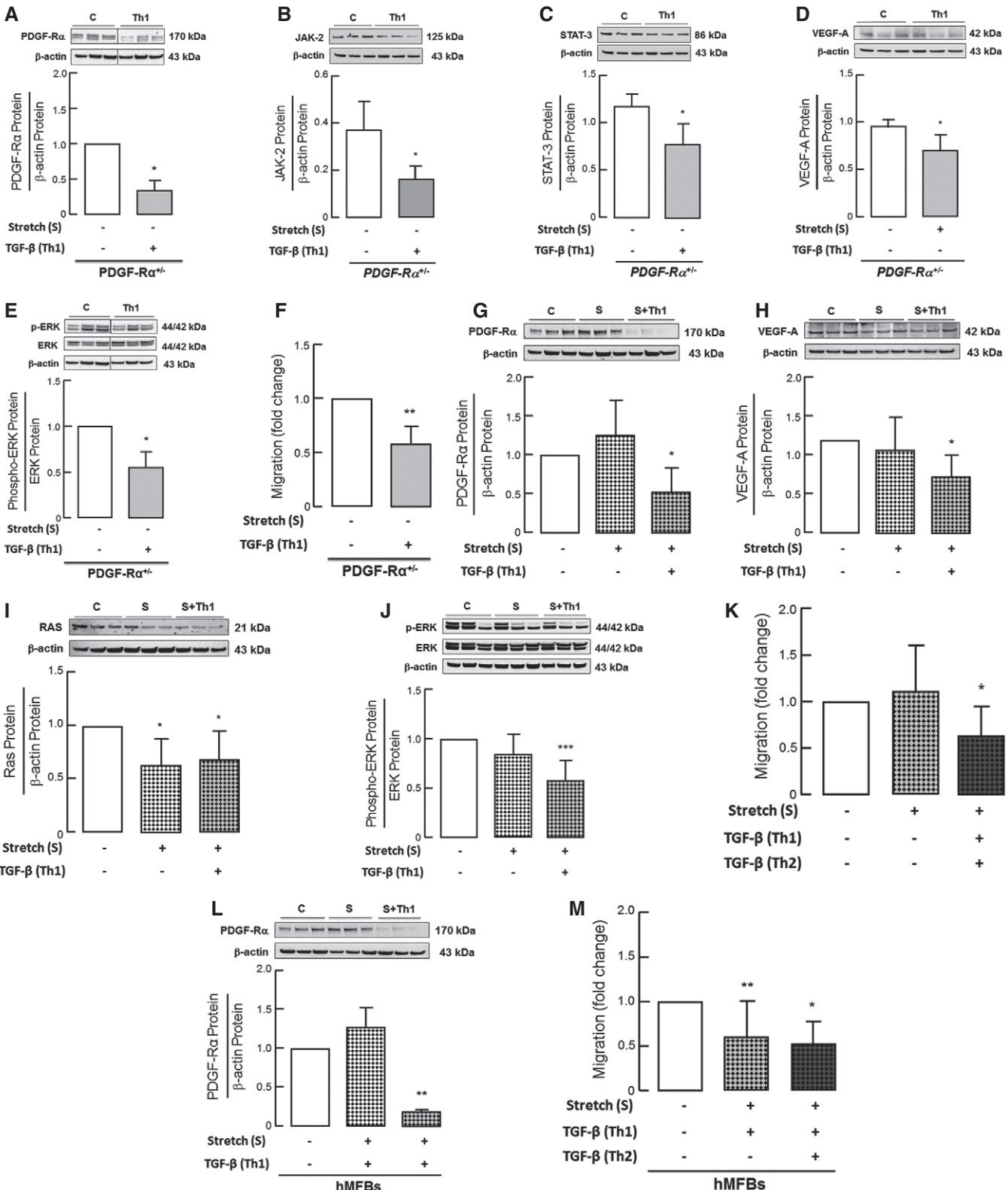

**Figure 6.**

We further confirmed the effect of TGF-β on PDGF-Rα signaling in primary pulmonary myofibroblasts isolated from WT neonatal mice and fibroblasts isolated from tracheal aspirates of nCLD patients. This analysis showed a significant downregulation in PDGF-Rα level, its signaling measure by pERK/ERK and impaired function displayed as reduced migration (Fig 5E, F and H). In accordance with the *in vivo*

◀

**Figure 6. Pronounced effect of TGF-β on pulmonary myofibroblasts from PDGF-Rα$^{+/-}$ mice and in concert with mechanical stretch on both mice and human myofibroblasts.**

A–E  TGF-β application (Th1) to myofibroblasts (MFBs) isolated from neonatal PDGF-Rα$^{+/-}$ mice reduced PDGF-Rα (A), JAK-2 (B), STAT-3 (C), VEGF-A (D), and pERK/ERK (E) protein levels when compared to control ($n$ = 3–6 mice/group). Panels (A, E) and (B, C) are from same blot hence having same β-actin bands.

F  Reduced migration assessed by Boyden chamber in myofibroblasts (MFBs) isolated from PDGF-Rα$^{+/-}$ mice compared to WT mice ($n$ = 5 mice/group).

G–J  TGF-β application in combination with mechanical stretch (S+Th1) in myofibroblasts (MFBs) isolated from neonatal WT mice showed reduced PDGF-Rα (G) and VEGF-A (H) protein levels as well as migratory RAS (I) and pERK/ERK (J) protein levels when compared to control myofibroblasts as assessed by immunoblot assay ($n$ = 6–9 mice/group).

K  TGF-β application (Th1) as an additional dose (Th2) on stretched myofibroblasts (MFBs) from WT mice reduced migration as assessed by Boyden chamber assay ($n$ = 5 mice/group).

L, M  TGF-β application in combination with stretch (S+Th1) reduced PDGF-Rα protein levels in fibroblasts (hMFBs) isolated from tracheal aspirates of nCLD patients when compared to control (C) or stretched (S) myofibroblasts (L) and as an additional dose (Th2) to stretched fibroblasts reduced migration (M) ($n$ = 3–6 patients/group).

Data information: Values are normalized to the respective controls except for (B–D). Data are presented as mean ± SD. Statistical test used in (A, D–F) is two-tailed and in (B, C) is one-tailed Student's $t$-test or Mann–Whitney test ($P$ = 0.004–0.05) and in (G–M) is ordinary one-way ANOVA with Bonferroni's correction ($P$ = 0.0001–0.04). ***$P$ < 0.001, **$P$ < 0.01, *$P$ < 0.05. C, un-stretched untreated control; S, stretched myofibroblasts (24 h); Th1, un-stretched myofibroblasts subjected to 5 ng/ml TGF-β (24 h); S+Th1, myofibroblasts stretched in parallel to TGF-β application (5 ng/ml) (24 h); Th2, re-incubation with 5 ng/ml TGF-β (8 h).

Source data are available online for this figure.

data, VEGF-A signaling associated with micro-vessel development was also diminished by 20% in pulmonary myofibroblasts from neonatal WT mice incubated with TGF-β (Fig 5G). Translating this finding in human, we demonstrate diminished PDGF-Rα levels and migration of fibroblasts (Fig 5I). Reduced migration upon TGF-β was confirmed by Boyden chamber and wound migration assays showing significantly abrogated velocity and distance travelled by fibroblasts in comparison to untreated controls (Fig 5J–M).

**Pronounced effect of TGF-β on pulmonary myofibroblasts from PDGF-Rα$^{+/-}$ mice and in concert with mechanical stretch on both mice and human myofibroblasts**

The inhibitory effect of TGF-β on PDGF-Rα level and downstream proteins JAK-2, STAT-3, and pERK/ERK was dramatic in myofibroblasts isolated from PDGF-Rα$^{+/-}$ mice with more than 30–50% reduction in the respective proteins (Fig 6A–C and E). This was accompanied by reduction in vascular marker VEGF-A and diminished migration (Fig 6D and F).

Mechanical ventilation has been demonstrated to exert significant strain forces on the developing lung in infants requiring invasive and even non-invasive respiratory support (Konig & Guy, 2014). We therefore dissected the contribution of mechanical stretch and the impact of growth factor exposure on PDGF signaling *in vitro*. Here we found that mechanical stretch in combination with TGF-β significantly reduced PDGF-Rα level in mouse myofibroblasts together with a matched reduction in VEGF-A expression confirming our *in vivo* findings (Fig 6G and H). With respect to myofibroblast function, we found that mechanical stretch in the presence of single or repeat doses of TGF-β significantly reduced migration in primary pulmonary myofibroblasts from WT mice together with a significant reduction in RAS and downstream pERK/ERK protein level, a signaling downstream of PDGF-Rα with a crucial role in myofibroblasts migration (Fig 6I–K; Liu *et al*, 2007; Fuentes-Calvo *et al*, 2013). In addition, mechanical stretch in the presence of TGF-β increased myofibroblast proliferation, together with proteins associated with proliferation PI3K and PCNA level (Fig EV2B–F). Whereas TGF-β in the absence of mechanical stretch was able to achieve comparable effects on myofibroblast migration, PDGF-Rα, and VEGF-A protein level (Fig 5E, G and H), mechanical stretch alone did not alter the migratory behavior or protein expression in the lung myofibroblasts

but did increase their proliferative behavior (Figs 6G and H, and EV2A and C).

Translating these effects to primary human fibroblasts obtained from tracheal aspirates of preterm nCLD patients, we found a significant reduction in PDGF-Rα protein level and migration by TGF-β in combination with stretch (Fig 6L and M). In line with this, repeat application of TGF-β in combination with stretch markedly reduced the migration of human lung fibroblasts (Fig 6M).

Taken together, these results indicate that elevated TGF-β signaling in the setting of MV-O$_2$ contributes to the development of nCLD, exacerbating the deficiency in PDGF signaling by inhibiting expression of PDGF-Rα on lung myofibroblasts.

The central role of the PDGF signaling cascade closely intertwined with upstream effectors and downstream effects with critical consequences for cellular functions in the injured neonatal lung as well as its recue with administration of PDGF-A is depicted in Fig 7.

# Discussion

Neonatal chronic lung disease (nCLD), formerly known as bronchopulmonary dysplasia, has long-term health consequences not only for pulmonary but also neurologic function. Invasive and non-invasive mechanical ventilation with oxygen-rich gas (MV-O$_2$) is necessary for the survival of preterm babies suffering from respiratory failure due to lung immaturity and insufficient respiratory drive after birth. Nonetheless, both treatments are known to contribute to adverse pulmonary outcome, that is, the development of nCLD (Konig & Guy, 2014). Hence, it is critical to pursue medical treatments that can prevent or treat nCLD in neonates requiring ventilatory support. Here, we provide causal evidence supporting a surprising cellular and molecular model for the development of nCLD. As shown in mouse and human, PDGF signaling not only affects secondary septation but significantly impacts the microvascular structure in close relation with the omnipresent TGF in the injured neonatal lung (Fig 7A). This close intertwinement of growth factor signaling in nCLD pathology with PDGF as a central driver holds promising potential for therapeutic approaches.

The integration of PDGF-Rα haploinsufficient mice with a unique preclinical model of nCLD, together with tailored biochemical and *in vitro* assays of human and mouse primary lung cells, allowed us

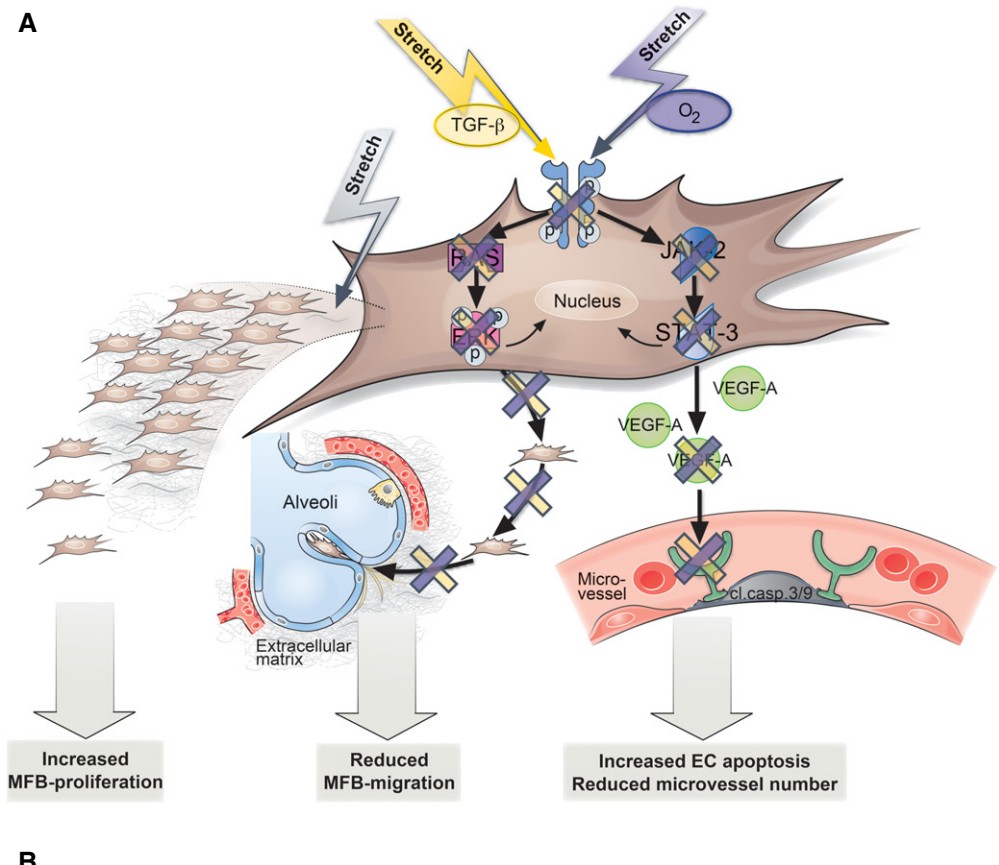

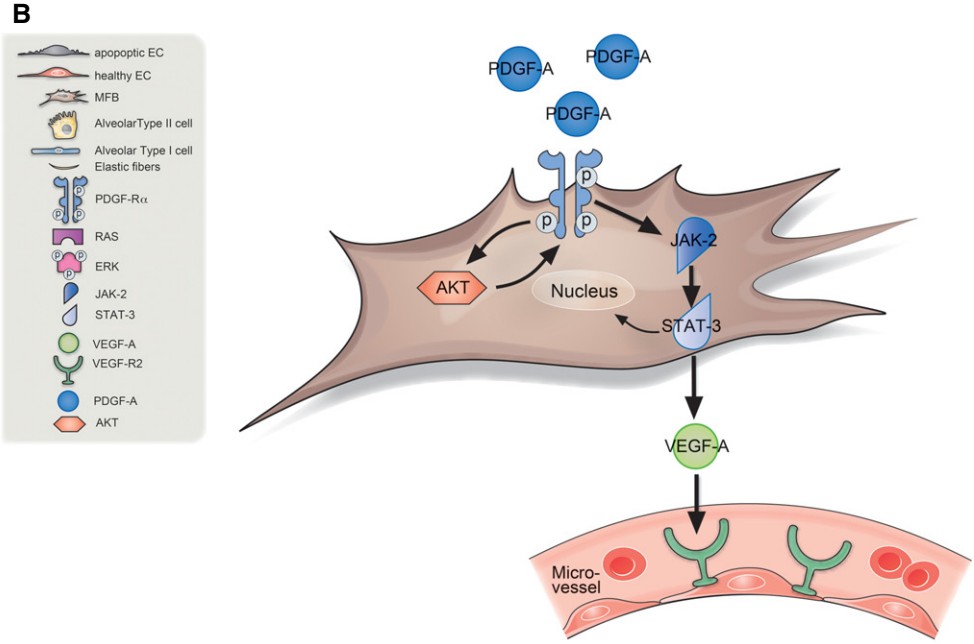

**Figure 7. Model for how attenuated PDGF signaling and positive pressure ventilation interact to produce the distinct phenotypic manifestations of nCLD.**

A   MV-$O_2$ *in vivo*, a combination of $O_2$, that is, oxygen and stretch (purple arrow) and/or TGF-β alone or in combination with mechanical stretch *in vitro* (yellow arrow), reduces platelet-derived growth factor receptor α (PDGF-Rα) levels and its downstream signaling through JAK-2 and STAT-3 in the pulmonary myofibroblast (MFB). This reduction in turn abrogates vascular endothelial growth factor expression (VEGF-A and VEGF-R2), leading to increased apoptosis in pulmonary endothelial cells (EC). Whereas myofibroblast migration is diminished through reduced RAS and pERK/ERK signaling, stretch alone increases their proliferation, hence depicting the differential effect of the most important denominators of nCLD development in the premature lung undergoing MV-$O_2$.

B   Application of PDGF-A to premature lung increases PDGF-Rα levels in an AKT-dependent manner in turn activating the downstream cascade through JAK-2, STAT-3 signaling. This then activates VEGF-A secretion and VEGF-R2 activity reducing apoptosis in endothelial cells (ECs).

to elucidate the molecular pathogenesis of this disease to an unprecedented level of understanding. Building upon previous observations suggesting a role for reduced PDGF-Rα signaling in lung pathology (Bostrom *et al*, 1996; Lindahl *et al*, 1997; Bostrom & Betsholtz, 2002; Lau *et al*, 2011; Chen *et al*, 2012; Popova *et al*, 2014), we dissected upstream and downstream molecular regulators of nCLD, including a crosstalk with VEGF-A and TGF-β with mechanical stretch (Fig 7A).

Our finding that impairment of the PDGF signaling pathway—previously associated only with myofibroblast migration and secondary septation of air sacs—is capable to produce both the alveolar structural and microvascular pathology of nCLD is surprising. Is it possible that due to the dynamic intercellular crosstalks required to generate the highly stereotyped and architecturally complex arrangement of alveoli, disruption of a single program indirectly disrupts closely coordinated, but distinct, processes? There is some precedent for this model, since inhibition of VEGF in lung development has been shown to result in reduced epithelial proliferation and impaired sacculation (Zhao *et al*, 2005). However, as we show in the case of PDGF-Rα, the causal relationship may be more direct, since VEGF-A is apparently produced not only by lung epithelial cells, but also by myofibroblasts in response to PDGF and its downstream signaling through JAK and STAT (Niu *et al*, 2002; Yu & Jove, 2004). Perhaps this additional level of patterned VEGF-A production is important for ensuring proper investment of newly forming secondary septal walls by the microvascular network. In any case, our demonstration of the therapeutic activity of exogenously administered PDGF-A (Fig 7B) for rescuing both the air sac septation and microvascular defects is highly promising, particularly since previous attempts to improve vascularization by administering exogenous VEGF-A not only failed to rescue nCLD, but actually induced capillary leakage (Akeson *et al*, 2005).

# Materials and Methods

All the antibodies used in the manuscript could be found in 1DegreeBio database. Study design for the manuscript experiments is described in Appendix Supplementary Methods.

## Human studies

### Patient characteristics

Two study cohorts comprising of 1,061 preterm infants at or less than 32 weeks of gestational age (GA) with and without nCLD, that is, BPD grade 2 or 3 according to Jobe & Bancalari (2001) [Pneumonia Research Network on Genetic Resistance and Susceptibility for the Evolution of Severe Sepsis (PROGRESS); German Neonatal Network (GNN)], were included in the SNP analysis (Ethics Approval #65/07, Homburg, University of Saarland; #145-07, Munich, Ludwig Maximillian's University of Munich and #File 79/01, Giessen, University of Giessen, Germany). Detailed patient characteristics are in Appendix Supplementary Methods. Nine patients out of this cohort were subjected to PDGF-Rα transcriptome analysis in association with the presence of single nucleotide polymorphism (SNPs). Patient characteristics of this cohort are in Appendix Table S3A. A separate study cohort was obtained from Perinatal Center of the Ludwig-Maximilians-University, Campus Grosshadern (Ethics Approval #195-07, Munich, Ludwig Maximillian's University of Munich, Germany) for SNP and protein analysis using SOMAscan ($n = 13$). The patient characteristics are in Appendix Table S3B. Tracheal aspirates of few patients from this patient cohort were used to isolate primary fibroblasts ($n = 6$). Patient characteristics of this cohort are in Appendix Table S3C. Sections from human lung were available through the Department of Pediatric Surgery at the Erasmus Medical Center. Lung samples were retrieved from the archives of the Department of Pathology of the Erasmus MC, Rotterdam, following approval by the Erasmus MC Medical Ethical Committee. Patient characteristics of this cohort are in Appendix Table S3D. According to Dutch law following consent to perform autopsy, no separate consent is needed from the parents to perform additional staining of tissues. A group of 20 preterm infants from Giessen was subjected to microarray analysis (Ethics Approval #File 79/01, Giessen, University of Giessen, Germany). The clinical course of all infants was comprehensively monitored. Approval of the local ethics committee and written informed parental consent was obtained for all samples studied. All the experiments conformed to the principles set out in the WMA Declaration of Helsinki and Department of Health and Human Services Belmont Report. The limitation of human material to be tested in case of neonatal chronic lung disease is well known; hence, analysis was performed with samples available in maximum capacity. Samples were collected from patients randomly. Our datasets were obtained from subjects who have consented to the use of their individual genetic data for biomedical research, but not for unlimited public data release. Therefore, we submitted it to the European Genome-phenome Archive (accession number—EGAS00001002586, study unique name—ena-STUDY-IMI-24-07-2017-10:03:30:362-576), through which researchers can apply for access of the raw data.

### SNP and protein analysis

Cord blood samples of 1,061 preterm infants at or below 32 weeks of gestational age ($n = 492$ BPD cases) were collected in ethylenediaminetetraacetic acid (EDTA) neonatal collection tubes. Genotypes of this patient cohort for PDGF-Rα SNPs (single nucleotide polymorphisms) were determined using Affymetrix Axiom microarrays based on the Axiom CEU array supplemented with some custom content. Matched controls were selected according to gender, GA, birth weight < $10^{th}$ percentile, and country of maternal origin. Case–control analysis adjusted for relatedness; 117 SNPs were measured in or near the PDGF-Rα. Whole-blood samples collected from a separate patient cohort ($n = 13$) that were analyzed for three significant SNPs by Eurofins Genomics were subjected to proteomic screening for PDGF-R and VEGF-A (SOMAscan™, SomaLogic, Boulder, USA). For a detailed description, please refer to Appendix Supplementary Methods.

### Human primary lung fibroblasts

Human primary lung fibroblasts were extracted from serial tracheal aspirate samples obtained from ventilated preterm infants (mean GA $24.9 \pm 1$ weeks, $n = 6$) later developing nCLD, at $4.7 \pm 1$ and $21.7 \pm 8$ day of life (Ethics Approval #195-07, Munich, Ludwig Maximillian's university of Munich, Germany). Human lung fibroblasts were cultured until 80% confluency in DMEM medium with 2 mM L-glutamine, Pen/Strep, and 20% FCS (PAN Biotech GmbH).

Purity of cell cultures was > 95% and FACS verified expression of CD11b (< 3%) (eBiosciences #48-0112-80), CD11c (< 3%) (BD Biosciences #557401), CD14 (< 5%) (BD Biosciences #09475A), CD45 (< 5%) (BD Biosciences #552848), CD90 (> 95%) (eBiosciences#48-0900-80), and CD105 (> 95%) (Miltenyi Biolabs #130-092-930); differences between patient samples were < 5% (Appendix Fig S3A). All fibroblast cultures expressed α-SMA as detected in cytosolic cell lysates by immunoblot analysis. This section is further described in Appendix Supplementary Methods.

### Gene expression microarray analysis

250–300 μl of whole cord blood was obtained from 20 preterm infants 72 h after birth and directly transferred to 750–900 μl of the PAXgene Blood RNA System (PreAnalytiX, Heidelberg, Germany). RNA was isolated using PAXgene Blood RNA System (PreAnalytiX) and was subjected to CodeLink Human Whole Genome Bioarrays (GE Healthcare). PDGF-Rα gene expression in GWAS patient cohort was measured by the expression of NM_006206 on the human whole genome bioarray and human 10 k I bioarray by Codelink. For details of blood sampling and RNA analysis, please refer to Appendix Supplementary Methods.

### Human lung slides

Human slides were obtained from paraformaldehyde-fixed and paraffin-embedded autopsy lungs from preterm infants with different BPD grades ($n = 7$) and an infant that died from a non-pulmonary cause. Tissue sections were stained for PDGF-Rα and TGF-β for further quantification. Please refer to Appendix Supplementary Methods for details of patient characteristics.

### *In vivo* studies

All the studies were performed as per ARRIVE guidelines.

### Gene-targeted mice

Gene-targeted mice (B6.129S4-Pdgfra$^{tm11(EGFP)Sor}$/J) referred to in manuscript as PDGF-Rα$^{+/-}$ or PDGF-Rα haploinsufficient mice were purchased from Jackson laboratories (Bar Harbor, ME, USA). Heterozygous mice are healthy and viable and reported to have no lung abnormalities (Hamilton et al, 2003). Co-staining for PDGF-Rα and α-SMA showed a reduction in myofibroblasts (double-positive), and mRNA and immunoblot analysis confirmed the reduced abundance of pulmonary PDGF-Rα expression in unventilated PDGF-Rα$^{+/-}$ mice when compared to WT littermates (Fig EV1A–C). For the study, 5–8-day-old neonatal mice both males and females were used. The experimental protocols were approved by the Bavarian government (TVA no. 55.2-1-54-2532-117-2010). The mice were kept under specified pathogen-free (SPF) conditions in a 12/12-h light cycle in the fully climate-controlled rooms having set points to the new conventions 2007/526 EC in our central mouse facility. Mice had a 2-week adaptation phase to their new environment and a handling by new nurses before putting them into the experiment. For each experiment, appropriate sample size ($n = 6–12$) was estimated considering the effect of MV-O$_2$ on viability of newborn mice.

### Mechanical ventilation

We used 5–8-day-old (newborn/neonatal) C57B6 wild-type (PDGF-Rα$^{+/+}$) and PDGF-Rα haploinsufficient (PDGF-Rα$^{+/-}$) mice, all born at term gestation weighing 4 g [WT $3.98 \pm 0.55$ g; PDGF-Rα$^{+/-}$ $3.93 \pm 0.66$ g bodyweight (bw)] to perform experiments in four groups of mice (14–16 mice per group). WT and PDGF-Rα$^{+/-}$ pups received mechanical ventilation with oxygen-rich gas (40% O$_2$) (MV-O$_2$) for 8 h at 180 breaths/min (MicroVent 848; Harvard Apparatus) after tracheotomy under sedation with ketamine and xylazine (60 and 12 μg/g bw), as previously described (Hilgendorff et al, 2011). The ventilator setting mimicked the clinical setting to avoid severe lung injury (mean tidal volume 8 μl/g bw; mean airway pressure (11–12 cmH$_2$O)). Tidal volumes were similar between the MV-O$_2$ groups (WT $8.3 \pm 0.5$ μl/g bw; PDGF-Rα$^{+/-}$ $7.9 \pm 0.3$ μl/g bw). The ventilation protocol was designed to avoid severe lung injury typically occurring in response to MV with very high inflation pressures and extreme hyperoxia. Hence, we used modest tidal volumes (mean 8.7 μl/g bw) and airway pressures (peak 12–13 cmH$_2$O, mean 11–12 cmH$_2$O), and limited the FiO$_2$ to 40%, thereby simulating the MV strategy of choice for preterm infants with respiratory failure. Respective controls spontaneously breathed 40% O$_2$ for 8 h after receiving sham surgery (superficial neck incision) under mild sedation. During MV-O$_2$, mice were maintained at neutral thermal environment; sedation with ketamine and xylazine (10 μg/g bw and 2 μg/g bw, respectively) was repeated as needed to minimize spontaneous movement and assure comfort. At the end of each study, pups were euthanized with an intraperitoneal overdose of sodium pentobarbital, ~150 μg/g bw, and lungs were excised for various studies as described below including histological analysis, as well as protein measurement and RNA expression analysis from frozen lung tissue. For PDGF-A treatment, 10 μl/g bw of sterile saline containing 25 ng/ml PDGF-A was administered through the endotracheal tube immediately before the onset of MV-O$_2$ as described previously. In a subgroup of mice with or without PDGF-A treatment, tidal volume and maximum tracheal pressure (Ptramax) was measured by whole-body plethysmography (Pulmodyn, Harvard Apparatus). Stepwise increase in the tidal volume allowed to assess quasi-static compliance (Hilgendorff et al, 2011).

All animals were viable with response to tactile stimulation and adequate perfusion at the end of each experiment. All surgical and animal care procedures were reviewed and approved by the Institutional Animal Care and Use Committee (Bavarian Government). Upon ventilation, both the mouse strains showed similar activation of pSMAD-2, while neonatal PDGF-Rα$^{+/-}$ mice displayed increased apoptosis when compared to wild-type mice (Fig EV3A–C). Allocation of the newborn mice to the groups was performed on the basis of similarity in weight and age. When newborn mice from more than one cage were used for the experiments, all pups were randomized before distribution in the groups. Investigators were blinded for the sex of mice during allocation. Further, the analysis of lungs was performed based on internal serial numbers allotted to mice blinding the investigator for genetic background and treatment (e.g., controls or ventilated). All experimental procedures were carried out in a laboratory-controlled environment during daytime.

### Quantitative histology and immunostaining

Lungs ($n = 6–11$/group) were fixed intra-tracheally with 4% paraformaldehyde overnight at 20 cmH$_2$O, as previously described (Bland et al, 2008). Fixed lungs were then excised, and their volume was measured by fluid displacement (Scherle, 1970). Lungs were embedded in paraffin for isotropic uniform random

(IUR) sectioning, as described previously (Scherle, 1970). Tissue sections (4 μm) were stained with hematoxylin and eosin (H&E) for quantitative assessment of alveolar area and number of incomplete and complete alveolar walls (septal density) in 2–3 independent random tissue sections per animal using the CAST image analysis system (CAST-Grid 2.1.5; Olympus, Ballerup, Denmark). Alveolar area, number of incomplete and complete alveolar walls (septal density), and radial alveolar counts providing an index of alveolar number were assessed. A minimum of 30 fields of view were quantitatively assessed in 2–3 independent random 4-μm H&E stained tissue sections per animal (CAST-Grid 2.1.5; Olympus; Emery & Mithal, 1960). Tissue sections were stained for PDGF-Rα, VEGF-A, cleaved caspase-3, CD31, and αSMA for further quantification (see Appendix Supplementary Methods for detailed description).

### Quantification of micro-vessels (20–100 μm)

20- to 100-μm-diameter blood vessels were assessed in H&E (normalized to 100 alveoli)- and CD-31-stained slides obtained from 8-h studies ($n = 6$–8/group) applying a previously described immunohistochemical and morphometric approach (Hilgendorff *et al*, 2011) in 30 fields of view in the distal lung/animal (400× magnification).

### Protein extraction and immunoblot analysis

After 8 h of MV-$O_2$, protein extraction from snap-frozen total lungs was done using high urea buffer ($KPO_4$, Urea, AppliChem) with Halt Protease Inhibitor (#1861280, Thermo Fisher Scientific). Primary lung fibroblasts were lysed with RIPA buffer with Halt Protease Inhibitor (mouse) or sodium vanadate (human) (#S6508, Sigma) and complete mini (#11836170001, Roche) followed by sonication. After measurement of protein concentrations (BCA, #23227, Pierce Scientific), immunoblots were performed using a Bis-Tris or a Tris-Acetate gel (#NP0321BOX, #EA0375BOX, Life Technologies) using the following antibodies: PDGF-Rα (C-20, Santa Cruz Biotechnology #338), VEGF-A (147, Santa Cruz Biotechnology #507), VEGF-R2 (Abcam, Cambridge, USA #Ab2349), VE-cadherin (H-72, Santa Cruz Biotechnology #28644), cleaved caspase-3 (Cell Signaling Technology #9661), cleaved caspase-9 (Cell Signaling Technologies #7237), eNOS (Cell Signaling Technologies #5880), phospho-ERK (Cell Signaling Technologies #4370), total ERK (Cell Signaling Technologies #4695), RAS (Cell Signaling Technologies #8955), PI3K (Cell Signaling Technologies #13666), JAK-2 (Cell Signaling Technologies #3230), STAT-3 (Cell Signaling Technologies #9139). Images were detected by chemiluminescence (#RPN2232, GE Healthcare) and quantified by densitometry (Bio Rad). Details of immunoblot analysis could be found in Appendix Supplementary Methods.

### *In vitro* experiments

### Mouse primary pulmonary myofibroblasts

Mouse myofibroblasts (MFBs) were extracted by excising lungs of 5–7-day-old C57BL6 wild-type mice after intraperitoneal overdose of sodium pentobarbital (~150 μg/g bodyweight). Under sterile conditions, lungs were flushed with PBS after cannulation of the right ventricle. Flushed lungs were then excised, diced into 1-mm pieces, and distributed on a petri dish (Corning #430167, Tewksbury, MA,

USA). Attachment to the dish was accomplished by incubation for 15–20 min at 37°C. Afterward, the tissue pieces were gently submerged in media (Gibco #41966-029, Darmstadt, Germany) containing Pen/Strep (Gibco, #15140-122) and Gentamycin (Lonza #BE02-012E, Basel, Switzerland) for 48 h before changing to fresh media. Experiments were started at 70–80% confluency. Myofibroblasts were characterized with fluorescence-activated cell sorter (FACS LSRII) using multicolor staining technique. Briefly myofibroblasts were resuspended in FACS buffer (PBS + 2% FCS + 10 mM HEPES + 0.1% Na-Azide) and stained with CD90.2 APC FITC (BD Pharmingen, Heidelberg, Germany #561974), CD 105 PE (BD Pharmingen #562759), CD45 FITC (BD Pharmingen #553080), CD11b V450 (BD horizon, Heidelberg, Germany #560455), CD11c PerCpCy5.5 (Biolegend, Fell, Germany #117328). For detection of internal markers, myofibroblasts were fixed with 4% PFA (Alfa Aesar GmbH, Germany #43368) followed by permeabilization with 0.2% Triton X-100 (Carl Roth GmbH + Co.KG, Karlsruhe, Germany #3051.2) and blocking with 1% BSA (Sigma) in PBS. Myofibroblasts were then stained in blocking solution with PDGF-Rα APC (eBiosciences #17-1401-81), α-Smooth Muscle Actin PE (R&D systems, Minneapolis, MN, USA #IC1420P), and Vimentin Alexa 488. Stained myofibroblasts were then acquired through BD™ LSR II utilizing BD FACSDiva™ software version 6.0 and analyzed using Flowjo version 9.6.1. As displayed in Appendix Fig S3B, myofibroblast culture constituted leukocytes ($0.6 \pm 0.5\%$ CD45$^+$), mesenchymal-like cells ($8.5 \pm 4.5\%$ CD105$^+$, $32 \pm 8.6\%$ CD90$^+$), and myofibroblasts ($77.2 \pm 14\%$ PDGF-Rα$^+$/Vimentin$^+$, $16.7 \pm 12\%$ Vimentin$^+$, and $77.6 \pm 27\%$ αSMA$^+$). Antibodies used were CD45 (BD Pharmingen #553080), CD105 (BD Pharmingen #562759), CD90 (BD Pharmingen #561974), PDGF-Rα (eBiosciences #17-1401), Vimentin (Cell Signaling #9854), and α-SMA (R&D systems #IC1420P).

### Mechanical stretch

Myofibroblasts were seeded on flexible-bottomed laminin-coated culture plates (#BF-3001L, Flex Cell International Corporation) and stretched at 70–80% confluency (shape/sine; elongation min 0%, max 8%; frequency 2 Hz; duty cycle 50%; cycles 43216) for 24 h. A detailed description of mechanical stretch experiment is available in the Appendix Supplementary Methods.

### Protein analysis

Cells were lysed with RIPA buffer including Halt Protease Inhibitor Cocktail (mouse myofibroblasts) or sodium vanadate (catalog #S6508, Sigma) and complete mini (Roche, Penzberg, Germany #11836170001; human lung fibroblasts). After storage at −80°C, cell lysates were sonicated and processed for immunoblot analysis (as described in "*in vivo*" methods).

### siRNA transfection of pulmonary mouse myofibroblasts

Primary neonatal mouse myofibroblasts were transfected with either 100 nM specific siRNA against PDGF-Rα (Santa Cruz Biotechnology, Inc., Germany #sc-29444) or 100 nM control siRNA B (Santa Cruz Biotechnology #sc-44230) suspended in TurboFect (Thermo Fisher Scientific, Waltham, MA, USA #R0531) or remained untreated controls in siRNA transfection media (Santa Cruz Biotechnology, Inc., #sc-36868). Transfection with siRNA was repeated after 17 h.

### Caspase activity and immunoblot analysis in human umbilical vein endothelial cells

Mycoplasma tested human umbilical vein endothelial cells (HUVECs; Commercially obtained from Lonza #CC-2935) were cultured in EBM-2 Basal Media (#00190860, #cc-4176, Lonza, mycoplasma free) with EGM-2 Single Quots supplements (Lonza #cc-4176) on 0.2% gelatin (Sigma Aldrich #G1393)-coated 96-well plates (5,000 cells/well) or 6-well plates (150,000 cells/well). After obtaining stable culture conditions, HUVECs were incubated with culture supernatants collected from three groups of mouse myofibroblasts: (i) untreated myofibroblasts (TurboFect), (ii) control siRNA-treated myofibroblasts, and (iii) myofibroblasts treated with siRNA against PDGF-Rα. For caspase activity assay, incubation with 40 μg/ml anti-VEGF (C-1) antibody (#sc-7269, Santa Cruz) served as a positive control. Caspase activity was assessed after 6 h using the Caspase-Glo 3/7 assay kit (#G8091, Promega). After 6 h of incubation, caspase activity was assessed using the Caspase-Glo 3/7 assay kit (Promega GmbH, Germany #G8091) according to the manufacturer's instructions. Cells plated in 6-well plate were lysed after 6-h incubation, and lysate was processed for immunoblot analysis with cleaved caspase-9 (Cell Signaling Technologies #9509) and eNOS (Cell Signaling Technologies #880) protein.

### Luciferase assay

PDGF-Rα promoter inserted in a reporter plasmid was transfected in CCL206 stimulated with TGF-β1. Luciferase activity was assessed the next day using dual luciferase assay. For generation of reporter, please refer to Appendix Supplementary Methods.

### Functional assays for myofibroblasts

Mouse and human myofibroblasts or lung fibroblasts were subjected to analysis of proliferation (Cell titer Glo assay and manual counting) and migration (Boyden chamber assay and scratch migration assay) followed by the application of stretch and TGF-β as shown in Appendix Fig S3C. Proliferation assay and scratch migration assay were performed after 48 h of application, while migration assay was done after 8 h. For detailed description of proliferation, Boyden chamber and scratch migration assays please refer to Appendix Supplementary Methods.

### Statistical analysis

All datasets are presented as mean ± SD. Statistical analysis was performed using Prism 5 and 6 software package (GraphPad, San Diego, CA, USA). Two-way analysis of variance (ANOVA) and *post hoc* test with Bonferroni correction were performed to compare controls and mechanically ventilated WT (PDGF-Rα$^{+/+}$) and haploinsufficient (PDGF-Rα$^{+/-}$) newborn mice. For *in vitro* experiments, one-way ANOVA and *post hoc* test with Bonferroni correction were performed to compare more than two groups of myofibroblasts (immunoblot, migration, and proliferation analysis). For analysis of caspase activity to HUVECs, nonparametric Kruskal–Wallis test was performed. To compare datasets from two groups of either WT or PDGF-Rα$^{+/-}$ mice (immunoblot, migration and proliferation analysis, reporter assays), parametric unpaired Student's *t*-test with Welch's correction or the nonparametric Mann–Whitney test (for datasets with a skewed distribution) was performed with two-tailed or one-tailed analysis. Test groups were always compared

to control group. Differences were considered statistically significant when the *P*-value was < 0.05. For microarray, data were analyzed in a target gene approach using the Pearson's correlation coefficient to correlate expression of PDGF-Rα and TGF-β in preterm infants with and without BPD. Correlation coefficients in preterm infants with and without BPD were tested using the R-packages *psych*. Statistical methods applied for the SNP analysis are outlined above.

**Expanded View** for this article is available online.

## Acknowledgements

We sincerely thank the patients and their families of the PROGRESS, PROTECT, and GNN study cohort for their significant contribution to the study by providing the samples. Funding: FöFoLe Grant RegNr.: 690, DFG Grant HI 1315/5-1, PROGRESS Study Group (BMBF Grant 01KI1010C, BMBF Grant 01KI1010I), Young Investigator Grant NWG VH-NG-829 by the Helmholtz Gemeinschaft and the Helmholtz Zentrum Muenchen, Germany.

## Author contributions

The conception and design of the manuscript was done by AH, TJD, and OE; data were acquired by PO, IT, MK, TP, KF, RJR, and DSM; followed by analysis and interpretation done by AH, TJD, HE, AW, PA, WG, LG, PO, TP, TR, DSM, and NJ; finally, the manuscript was drafted for important intellectual content by AH, TJD, OE, AS, PO, and TP.

## Conflict of interest

The authors declare that they have no conflict of interest.

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
