## [Review Process File · EMBO Molecular Medicine]

Attenuated PDGF signaling drives alveolar and microvascular defects in neonatal chronic lung disease

Prajakta Oak, Tina Pritzke, Isabella Thiel, Markus Koschlig, Daphne S. Mous, Anita Windhorst, Noopur Jain, Oliver Eickelberg, Kai Foerster, Andreas Schulze, Wolfgang Goepel, Tobias Reicherzer, Harald Ehrhardt, Robbert J. Rottier, Peter Ahnert, Ludwig Gortner, Tushar J. Desai, and Anne Hilgendorff

*Corresponding authors: Tushar Desai, Stanford University School of Medicine
Anne Hilgendorff, Ludwig-Maximilian University of Munich*

Review timeline:	Submission date:	14 November 2016
	Editorial Decision:	19 December 2016
	Revision received:	22 March 2017
	Editorial Decision:	12 April 2017
	Revision received:	25 July 2017
	Accepted:	27 July 2017

Transaction Report:

Editor: Roberto Buccione

1st Editorial Decision

19 December 2016

Thank you for the submission of your manuscript to EMBO Molecular Medicine. We have now heard back from the Reviewers whom we asked to evaluate your manuscript.

We are sorry that it has taken longer than usual to get back to you on your manuscript. In this case we experienced difficulties in securing appropriate reviewers and then obtaining their evaluations in a timely manner. Further to this, I wished to discuss the evaluations further with my colleagues

Reviewer 1 questions the news value of the work presented, while at the same time and similarly to the other reviewers, praising the quality of the work. S/he also notes the insufficient experimental support for some of the mechanistic aspects (e.g. how reduced PDGF signaling leads to effects on VEGFA and endothelial apoptosis, and how PDGFA increases PDFGRa). Reviewer 1 does however find the SNP data to be especially interesting, provided that these are linked to gene expression and PDGFR expression and PDGF signaling in clinical samples.

Reviewer 2, while more positive, does however raise a number of concerns. Among these and similarly to reviewer 1, insufficient mechanistic insight including on the role of TGFbeta signaling in the effects of reduced PDGF-signaling in nCLD, and, again, on the effects of PDGFA administration. Reviewer 2 also suggests a number of experiments to support and consolidate the claims but also note a number of shortcomings that need to be addressed to bring the manuscript to a sufficiently high qualitative standard in terms of ensuring reproducibility (a topic close to our hearts

at EMBO Molecular Medicine).

Finally, reviewer 3, perhaps the most reserved, questions the manuscript's overall relevance for human.

The three reviewers also list other items for your action.

After reviewer cross-commenting and further discussion, we agreed that you should be allowed to submit a substantially revised manuscript, with the understanding that the reviewers' concerns must be addressed with additional experimental data where appropriate and that acceptance of the manuscript will entail a second round of review. A clear consensus emerged to the effect that clear demonstration human relevance is crucial. I would like to make it clear that, lacking the above, a study on the PDGF pathway in alveolar simplification in mice would not be suited for publication in EMBO Molecular Medicine.

It is important that you consider that it is EMBO Molecular Medicine policy to allow a single round of revision only and that, therefore, acceptance or rejection of the manuscript will depend on the completeness of your responses included in the next, final version of the manuscript.

As you know, EMBO Molecular Medicine has a "scooping protection" policy, whereby similar findings that are published by others during review or revision are not a criterion for rejection. However, I do ask you to get in touch with us after three months if you have not completed your revision, to update us on the status. Please also contact us as soon as possible if similar work is published elsewhere.

Finally, please note that EMBO Molecular Medicine now requires a complete author checklist (<http://embomolmed.embopress.org/authorguide#editorial3>) to be submitted with all revised manuscripts. Provision of the author checklist is mandatory at revision stage; The checklist is designed to enhance and standardize reporting of key information in research papers and to support reanalysis and repetition of experiments by the community. The list covers key information for figure panels and captions and focuses on statistics, the reporting of reagents, animal models and human subject-derived data, as well as guidance to optimise data accessibility.

I understand that if you do not have the required data available at least in part, to address the above, this might entail a significant amount of time, additional work and experimentation and might be technically challenging, I would therefore understand if you chose to rather seek publication elsewhere at this stage. Should you do so, we would welcome a message to this effect.

Please note that we now mandate that all corresponding authors list an ORCID digital identifier. You may do so through our web platform upon submission and the procedure takes <90 seconds to complete. We also encourage co-authors to supply an ORCID identifier, which will be linked to their name for unambiguous name identification.

I look forward to seeing a revised form of your manuscript.

***** Reviewer's comments *****

Referee #1 (Comments on Novelty/Model System):

The author's provide well-integrated data from a series of in vitro, in vivo and human studies. There are however, significant limitation of the data, as outlined in my review, which limit the novelty and potential impact of this work.

Referee #1 (Remarks):

In this manuscript the authors report the importance of platelet-derived growth factor (PDGF) signaling in development of neonatal chronic lung disease (nCLD). They suggest that attenuated PDGF signaling is an important driver of nCLD both in animal models as well as in infants who require positive pressure ventilation and that PDGF-A may be a viable therapeutic option in patients.

The base their conclusion integration of data derived from in vitro, in vivo and patient experiments/data. Specifically they show that infants with nCLD have enrichment of SNPs in the PDGF-R α gene. Next they show that attenuated PDGF (PDGF-R α +/- mice) signaling in the setting of mechanical ventilation and O₂ supplementation (MV-O₂) independently contributes not only to impaired alveolar septation but also the microvascular defects of nCLD, likely by reduced VEGF-A expression in myofibroblasts. Further, they found elevated TGF- β expression in lungs of human infants undergoing ventilation. Further they show in vitro that TGF- β directly downregulates PDGF-R α expression and inhibits lung myofibroblast migration in concert with mechanical stretch. They make the case that PDGF mediates its effect through a myofibroblast specific mechanism. Lastly they show that exogenous PDGF-A rescued both the microvascular and air sac defects in haploinsufficient mice undergoing MV-O₂.

The experiments are technically well done and provide evidence for the importance of PDGF in the pathophysiology of nCLD. There is well integrated in vitro, in vivo and human data supporting their hypothesis. However the author's supposition that PDGF signaling is central to diminished alveologenesis or nCLD via a myofibroblast dependent mechanism is not novel and consistent with essentially downstream effects due to TGF- β . It has been previously shown in several publications, a few of which are referenced in this paper, that absence of PDGF-A or PDGF-R is associated with impaired alveologenesis. TGF- β has been shown previously to be central to the development of nCLD which is also supported the data in this paper. Further though TGF- β downregulates PDGF-R α , it has many other effects, including myofibroblast migration, that may be independent of PDGF signaling and. Thus it's not clear if the reduction of PDGF-R α serves as a biomarker for TGF- β mediated effects or is a central mediator of its effects. The rescue data with PDGF-A is interesting, but potentially supraphysiologic dosing of a compound does not make a convincing case for a central mediator. Thus the novelty of this approach is limited. Potentially the most important data relates to the SNPs. However no data is offered with respect to the impact of SNPs on gene expression, the relative abundance in the general population or whether the SNPs are associated with differential expression of genes in the PDGF pathway. I also have specific comments, listed below.

Specific Comments

- What is the relationship between PDGF-R α SNPs and gene expression?
- What is the prevalence of each SNP in the general population?
- Was the presence of a SNP associated with decreased expression of PDGF-R α or attenuated activation of the PDGF pathway in patient cells?
- The haploinsufficient mouse is not a tissue-restricted model so it is not convincing that the effects are mediated only or even primarily by myofibroblasts?
- The data in figure 3 would be more convincing if co-localization of VEGF-R2 and VE-cadherin with an endothelial marker was demonstrated.
- The authors provide no mechanism by which reduced PDGF signaling leads VEGF-A expression and endothelial apoptosis.
- The author's postulate that administration of PDGF-A increases PDGF-R α in a feed forward mechanism. Is there any data to support this? This is important to in order to validate use of PDGF-A as a therapy. It may be useful to assess global and cell specific gene expression in these experiments.
- It is not clear why lung volumes were not impacted if pulmonary compliance improved with PDGF-A treatment, since the two should be related.

Referee #2 (Remarks):

The authors investigated the role of PDGF-signaling in neonatal Chronic Lung Disease (nCLD) using different experimental system, such as samples from ventilated human newborns, samples from transgenic mice with PDGF-R α haplo-insufficiency (PDGF-R α +/-) and preclinical mouse model of mechanical ventilation with oxygen-rich gas (MV-O₂). Oak et al. suggest that PDGF-R α is a crucial molecular driver necessary for alveolar septation and vasculature integrity. In line with their model, exogenous PDGF-A rescues both structural and functional defects in the different experimental systems used for analysis of nCLD. Further, the authors show a decrease of PDGF-R α in some of the nCLD experimental systems, which correlates with an increase of TGF β -signaling, thereby suggesting a counteracting causal involvement of both signaling pathways. In addition, the authors suggest a protective role of PDGF-R α for infants that develop nCLD. Since nCLD is the

most common complication of preterm birth, the potential for clinical application of this work is compelling. Furthermore, even if a correlation between PDGF-signaling and nCLD has been previously reported (PMID: 24907056), the work presented by Oak et al. is conceptually novel because it explores in detail upstream- and downstream-effectors mediating the attenuation of PDGF-signaling in nCLD. Of special interest is the counteracting effect of PDGF- and TGFB-signaling in the context of nCLD. Due to the potential clinical application and the conceptual novelty of the work, I would like to suggest this manuscript for publication at EMBO molecular medicine (EMM) after major revision. In the following lines, I will elaborate my concerns and suggestions to the authors not only to reach the standards of publication at EMM, but also to strengthen the experimental evidences supporting their model thereby increasing the impact of the work in the scientific community:

Major concerns:

1. The authors should include a supplementary table with the clinical relevant data of patients with nCLD analyzed in the manuscript. For instances, gender, gestational age (av.), weight (av.), country of maternal origin, ventilation and O2 days (av.), severity of disease, survival, etc. This information will be useful for other scientists in the future implementing the results by Oak et al. in to their work.

The authors should write into the manuscript the number of approval by the ethic committee for the use of human samples for scientific purposes.

2. The authors should deposit the array-based transcriptome analysis into a public accessible data base, for instances at the NCBI. During the review process, they should provide "reviewers-only" access in order that the quality of the data can be determined before publication.

In addition, the results from the array-based transcriptome analysis should be confirmed by qRT-PCR based expression analysis of several of the target genes of the PDGF-signaling pathway. Perhaps some of the nominally associated genes presented in fourth column of the Suppl. Table 2 for the analysis of SNPs in PDGF-related pathways could be good options for the suggested qRT-PCR based expression analysis.

3. The experimental evidences presented by Oak et al. supporting the causal involvement of enhanced TGFB-signaling mediating the effects caused by reduced (or attenuated) PDGF-signaling during nCLD are relatively weak. Nevertheless, this is one of the attractive mechanistic aspects of the model proposed by the authors. The authors could improve this weak point performing the following experiments:

3.1 In Western Blot analysis (WB) of protein extracts after administration of PDGF-A in the Figure 4D, the authors should monitor the level of activity of TGFB-signaling. For example using antibodies (Ab) specific against phosphorylated SMAD proteins and/or against proteins encoded by genes that are known targets of TGFB-signaling. Based on the author's model, the administration of PDGF-A should reduce TGFB-signaling.

3.2 The inhibition of TGFB-signaling in similar experimental settings as presented in the Figure 4 should have similar effects as the administration of PDGF-A, thereby supporting the model presented by the authors. There is several ways to inhibit the TGFB-signaling. The author should select one or two ways to reduce (or block) TGFB-signaling in the experimental settings presented in the Figure 4.

In the same line of ideas, the most elegant and striking demonstration of the causal involvement of TGFB-signaling mediating the effects caused by attenuated PDGF-signaling during nCLD would be a rescue of the phenotype observed in the transgenic PDGF-R α ^{+/-} mice by crossing this line with a second transgenic mice line, in which the TGFB-signaling is attenuated. However, I recognize that the generation of this double-transgenic mice line will require additional time and efforts that are out of the scope of the present manuscript.

3.3 In addition to the immunostaining presented in the Figure 5A, the authors should monitor the activation of TGFB-signaling by WB analysis of protein extracts using Ab specific against pSMAD2. The possibility of determining the MW during WB analysis will further support the specificity of the results obtained using this Ab. In addition, comparisons of the levels of proteins between different samples are more accurate by WB than by immunostaining.

3.4 In the Figure 5D, the authors present a LUC-assay using a reporter-constructs containing part of the PDGF-R α promoter. First, the author should specify the region of the promoter cloned into the reporter plasmid. Second, a qRT-PCR based expression analysis of the endogenous PDGF-R α gene without or with TGF β -treatment will be more relevant than the presented LUC-assay.

In addition, I strongly disagree with the description of the results obtained in this LUC-assay. The authors write: "...To test for a direct interaction between TGF- β and PDGF-R α , we conducted a luciferase assay by transfecting CCL206 cells with a pGal vector carrying a PDGF-R α promoter insert. Administration of TGF- β caused a 50% reduction in luciferase activity (Fig. 5D), indicating that TGF- β directly inhibits PDGF-R α gene transcription, and supporting that the inverse relationship observed in nCLD is causal...". A direct interaction between two proteins could be demonstrated by Co-immunoprecipitation assays, protein pull down experiments, proximity ligation assays, among others, but not with the LUC-assay presented by the authors. Further, the direct inhibition of a gene could be demonstrated by Chromatin immunoprecipitation (ChIP) assays, electrophoretic mobility shift assays (EMSA) in combination with expression analysis. I recommend the authors to describe this experiment in an appropriate manner.

4. In Fig. 4E-F there is a dramatic increase of VE-Cadherin and VEGF-A upon PDGF-A treatment. The authors should evaluate the angiogenesis process in the lungs of ventilated mice, perhaps by endothelial cell tube formation assay or BrdU incorporation. The results will further support the model presented by the authors in the Figure 7, in which VEGF-A signaling is downstream of PDGF-signaling.

The author should include into the model presented in the Figure 7 the proposed effect of TGF β -signaling.

The author should describe the model presented in the Figure 7 at the beginning of the Discussion section. The last sentence in the current Results section could be used as introductory sentence for the description of the model in the Discussion.

5. By comparing the positions on the chromosomes of the SNPs in Fig. 1A (n=117) with the position on the chromosomes of annotated SNPs from previously published data sets using the Genome Browser and the human genome hg19 (n=147), I detected not only differences in the numbers of SNPs (n=117 vs. n=147), but also in their median distribution (the majority of peaks is located nearer PDGFRA 5' UTR). The authors should discuss the possible reasons for the observed discrepancies. In addition, the authors should explain why they used the human genome hg18 instead of hg19.

6. The colors for all the figures should be selected in order that "color-blind" individuals can still interpret the information. The combination of green and red in the immunostaining is not appropriate for color-blindness. They authors should refer to literature dealing with color combinations appropriate for color-blind individuals.

The colors for the Figure 1 A should be selected in a similar manner and in order that the different groups can be recognized easier.

7. The authors should use scale bars in all microscopy pictures (including the immunostainings) and describe them in the legend.

8. The author should increase the size of the pictures from the WB presented in all the figures starting from Figure 2 onwards. If there are complications due to limited space in the figures, the authors could consider presenting several WB pictures together in the same panel. For example, the WB pictures presented in the panels 2B and 3B-D could be presented together, since they are from the same experiment using different Abs during the immunodetection after WB. In a similar manner, the WB in the panels 3H and 3J could be presented together. The WB in the panels 4D-F could be presented together. The WB in the panels 5E-G could be presented together.

If the results are robust, I rather prefer to see the WB pictures than the densitometry analysis.

In addition, the pictures of the WB selected for the figure should correlate with the densitometry analysis. This is not always the case in the current manuscript. For example, in the Figure 2G the mean of the WT group in the plot is above 1.0, what would mean that the intensity of ACTB bands is lower than the intensity of the PDGF-R α bands. The opposite is the case in the pictures used for the Figure 2G.

9. The authors have to improve the description of the statistical analysis performed. The way how is described in the legends and in the Material and Methods section of the current version of the manuscript does not allow the reader to recognize which analysis was performed in each panel to determine the statistical significance of the values presented. In addition, the authors should specify the method used to determine the distribution of the data set.

Minor concerns:

10. The author should specify the company, catalog number of the Abs. used for immunostaining. The author should also specify the same information for the PDGF-A protein used in the manuscript.

11. The authors should consistently use in the manuscript the official names and official symbols for the proteins and the genes described, which can be found at NCBI. In addition, the authors should use the nomenclature standards for the description of proteins and genes in human and mouse. The research community has adopted and nomenclature standards for each model organism and published them on the relevant model organism websites and in scientific journals.

12. When referring to proteins and specially to post-translational modified proteins as phosphorylated SMAD or ERK proteins, I strongly suggest the authors to substitute the word "expression" by "levels" or "protein levels".

13. There are several inconsistencies in the legends of the figures. For example in Fig. 1 the legends of panels A and B are swapped. In Supp. Fig. 1 there is not Panel C, as it is referred in Methods "Gene-targeted mice".

14. There are some typos in the text. I suggest the authors to run once again the spelling and grammar command from the text program used to detect them. "Messages" should be substituted by "messengers" or "mRNAs", for instances in the Introduction, page 7. "mics" should be substitute by "mice" in the legend of the Figure 2C. The word "interaction" should be substituted by "cross-talk" when referring to signaling pathways. Consider rephrasing the titles in the Result section. For instances, the current title "Enrichment of PDGFR- α SNPs and reduced expression of PDGFR- α by human lung fibroblasts in ventilated preterm infants developing nCLD" could be substituted by "Enrichment of PDGFR- α SNPs and reduced expression of PDGFR- α IN human lung fibroblasts FROM ventilated preterm infants developing nCLD"

I hope that my suggestion support the authors during the revision of their manuscript.

Referee #3 (Comments on Novelty/Model System):

The results are quite consistent with their hypothesis in the mouse system. The challenge with this is that it is becoming increasingly clear that the mouse and human systems are not exactly the same. Thus the relevance of the very detailed findings in this paper are open to some question. This is a fairly recent development however. But I think it should at least be discussed.

Referee #3 (Remarks):

This is an interesting packet of information looking at PDGF in alveolar simplification from several angles. First they show that there are some SNPs in the PDGF pathway that may be relevant. Then they look at the response of PDGF pathway haplotypes to oxygen and ventilation. Then they look at the intercurrent role of TGFbeta pathway signaling in isolated myofibroblasts. Lastly they show a rescue study with PDGF ligand. This is a nice overall package although the way it is laid out is more confusing than it seems from what I have written here. So the manuscript could do with shortening and omg elf the data taking out as there are really several separate stories here I think.

The other comment is that it is not clear to me what relevant e these studies may have to human other than the SNPs because it is becoming increasingly clear that mouse models of lung development and injury repair, although they are very interesting and convenient may not actually have a lot to do with the fine details of human development and repair.

I also think that PDG ligand rescue in human would be very tricky. Mouse studies of excess PDGF ligand are far from reassuring that this will result in normal lung development in mice so that getting the dose response and administration regimens right will be hugely challenging. Nevertheless as a suite of models the paper clearly has some merit but maybe the ideas and claims of potential primary driving of the biological process and therapeutic efficacy should not yet be posited in a general interest journal.

1st Revision - authors' response

22 March 2017

Response to reviewers

We would like to thank the reviewers for their thoughtful and detailed critiques. We now submit a substantially revised manuscript incorporating a large amount of new experimental data that addresses the most significant concerns, namely, human relevance, the putative central role of PDGF signaling, and deeper dissection of PDGF-R α signaling mechanisms.

First, we focused on determining whether and how the implicated human SNPs affect the PDGF-R α gene. We therefore directly tested the consequences of the identified SNPs on gene expression and myofibroblast (MFB) function in humans. The results show reduced PDGF-R α mRNA, reduced PDGF-R α protein, and a functional impairment in migration of MFBs isolated from a new cohort of preterm infants harboring the SNP. These new human SNP analysis data are presented in **Figure 1C-G**, **Appendix Figure S1**, and **Appendix. Tables 1 and 2**. The new data on functional consequences of the SNP on human MFB activity are presented in **Figure 1G**, **Figure 6L-M**, and **Expanded view Figure EV2**. Overall, these new findings not only corroborate the human relevance of our mouse studies, but also strongly support that attenuated PDGF signaling plays a central role in the pathogenesis of nCLD, as discussed further below.

Next, we delved deeper into the specific molecular mechanisms by which attenuated PDGF signaling in MFBs contributes to nCLD. We find that PDGF signaling promotes increased expression of PDGF-R α via AKT and drives production of VEGF via the JAK/STAT3 pathways. We also find that conditioned medium from PDGF-R α siRNA-treated MFBs induces endothelial cell apoptosis, activating Caspase 3 and 9 and reducing eNOS. These new mechanistic data are presented in **Figure 2H-I**, **Figure 3J-K**, **Figure 4E-F, I**, **Figure 6B-C**, and are summarized in a new panel in our proposed model schematic (**Figure 7**). The findings flesh out the specific molecular mechanisms by which attenuated PDGF signaling in MFBs induces endothelial cell apoptosis and how signaling results in increased expression of PDGF-R α . Equally importantly, they corroborate the central importance of PDGF signaling in nCLD by showing how its attenuation independently promotes the air sac (reduced expression of PDGF-R α through AKT driving impaired alveolarization) and vascular (reduced production of VEGF through JAK/STAT3 and production of a secreted factor driving endothelial cell apoptosis) pathology.

Finally, we experimentally strengthened and more clearly articulated why we consider PDGF signaling to be a central driver of nCLD pathology. Most significantly, our new mechanistic analysis linking the implicated SNP to reduced PDGF-R α expression in human MFBs provides strong support that attenuated signaling is causally sufficient for nCLD, and not merely a biomarker or downstream effector of disrupted TGF- β signaling. We have also added new data indicating that the levels of TGF- β activation in WT and PDGF-R α haploinsufficient mice are comparable, presented in **Expanded view Figure EV3**. Furthermore our new mechanistic data on downstream pathways regulated by PDGF-R α activation (see above) have identified independent molecular mechanisms by which attenuated signaling drives the air sac and vascularization defects, the hallmarks of nCLD pathology, providing additional support for its central role rather than one of multiple “effector” arms of disrupted TGF- β signaling.

In summary, our new data making the link between the implicated SNP and reduced PDGF-R α expression, confirmation of reduced PDGF-R α production in human MFBs carrying the SNP, and demonstration that these fibroblasts have migration defects provides direct experimental proof implicating attenuated PDGF signaling in the pathogenesis of nCLD. This confirmation of human relevance, in turn, corroborates the significance of our mechanistic studies in the PDGF-R α haploinsufficient mouse model. By the same token, considering the therapeutic effect of exogenous

PDGF in our mouse model, it provides a rationale for pursuing strategies for augmenting PDGF signaling as a potential therapy in preterm infants for treating nCLD.

Below, we provide a point-by-point response to the specific concerns of each reviewer.

Referee #1:

The experiments are technically well done and provide evidence for the importance of PDGF in the pathophysiology of nCLD. There is well integrated in vitro, in vivo and human data supporting their hypothesis. However the author's supposition that PDGF signaling is central to diminished alveologenesis or nCLD via a myofibroblast dependent mechanism is not novel and consistent with essentially downstream effects due to TGF- β ². It has been previously shown in several publications, a few of which are referenced in this paper, that absence of PDGF-A or PDGF-R is associated with impaired alveologenesis.

We thank the reviewer for praising the quality of our work and our demonstration that PDGF is important for nCLD. As the reviewer points out, it is well-established that PDGF signaling is important for proper myofibroblast migration during alveolar morphogenesis, so it is not surprising that it also plays a role in nCLD. However, what is novel and unexpected is that attenuated PDGF signaling mediates the diverse aspects of nCLD pathology, including non-canonical microvascular defects that are not due to impaired MFB migration. We have also gone on to uncover the mechanism for its effect on endothelial cells, showing that MFBs with attenuated PDGF signaling downregulate VEGF production via the JAK/STAT pathway.

TGF- β ² has been shown previously to be central to the development of nCLD which is also supported the data in this paper. Further though TGF- β ² downregulates PDGF-R α ; it has many other effects, including myofibroblast migration, that may be independent of PDGF signaling and. Thus it's not clear if the reduction of PDFG-R α ; serves as a biomarker for TGF- β ² mediated effects or is a central mediator of its effects.

We appreciate the challenge of accurately inferring genetic hierarchies in conditions like nCLD, where multiple pathways are dysregulated, any of which could represent truncal (“driver”), downstream (“effector”), incidental (“passenger”) or compensatory (“responder”) programs. The reason we propose PDGF signaling is a central mediator is because 1) the implicated SNP reduces PDGF signaling and 2) attenuated PDGF signaling is sufficient to recapitulate the defining pathologic features of nCLD. Therefore, since it is both causally implicated and sufficient to produce nCLD, the reduction in PDGF-R α fits best in the “driver” or central mediator position of the genetic hierarchy. More specifically, we demonstrate that the reduction of VEGF signaling and the consecutive development of a vascular pathology are PDGF dependent as i) the pulmonary expression levels for pSMAD 2/3 are similar in both ventilated wildtype as well as PDGF-R α haploinsufficient mice (**Expanded view Figure EV3A and B**) and ii) In spite of pSMAD levels being same in both mice systems upon ventilation, the apoptosis as shown by cleaved caspase 3 expression is significantly more in PDGF-R α haploinsufficient when compared to wildtype mice (**Expanded view Figure EV3C**).

The rescue data with PDGF-A is interesting, but potentially supraphysiologic dosing of a compound does not make a convincing case for a central mediator. Thus the novelty of this approach is limited.

We appreciate the comments of the reviewer and agree that dosing is an important issue when designing treatment strategies. It is however totally unclear what physiologic and supraphysiologic doses are, especially under disease conditions. A dose-defining study would have to be performed, along with a careful assessment for any potential toxicity, before a clinical trial is undertaken. Whether or not PDGF signaling is considered a central mediator of nCLD, the observation that its administration can ameliorate both the air sac and microvascular defects is exciting because it provides an empiric rationale for pursuing augmentation of PDGF signaling as a potential therapy.

What is the relationship between PDGF-R-a SNPs and gene expression?

We now show a correlation between the presence of all 3 significant SNPs and reduced PDGF-R α expression in the cord blood samples of preterm infants (**Figure 1C and Appendix FigureS1**).

What is the prevalence of each SNP in the general population?

Because we analyzed the significance with respect to BPD or no BPD in the cohort with equal risks, i.e. preterm infants < 32 weeks GA, the prevalence in the healthy population cannot be related to our data. As there would be no control for stratification one would expect to see a trend, if population data were matched well enough but looking into a larger cohort – as data are not commonly available on that for all three or all 117 SNPs – we opted to not undertake this work with no direct application of the data to our work presented.

Was the presence of a SNP associated with decreased expression of PDGF-R α or attenuated activation of the PDGF pathway in patient cells?

Yes. **Figure 1F-G** shows reduced PDGF-R α expression and migration in fibroblasts isolated from tracheal aspirates of ventilated preterm infants homozygous for SNP rs12506783 versus fibroblasts heterozygous for the SNP.

The haploinsufficient mouse is not a tissue-restricted model so it is not convincing that the effects are mediated only or even primarily by myofibroblasts?

Our experimental assays identified specific defects in lung MFBs that support the local development of pathology (i.e., lung MFB migration for alveolarization and production of VEGF to signal to nearby lung endothelial cells, **Figure 2E-F, 3**). Furthermore, lung-directed administration of PDGF ligand was active in ameliorating the nCLD pathology, suggesting involvement of lung MFBs in disease pathogenesis (**Figure 4**). Finally, the lack of tissue-restriction in our mouse model mirrors the situation in the infants with nCLD who carry the implicated SNPs (i.e., it is a germline mutation that does not affect viability), so we do not feel it is imperative to perform a lung-restricted haploinsufficient mouse model. nCLD also manifests with non-pulmonary localized consequences (e.g., neurological), which may be due to PDGF-R α -expressing cells in the brain or other relevant tissues.

The data in figure 3 would be more convincing if co-localization of VEGF-R2 and VE-cadherin with an endothelial marker was demonstrated.

We now show in **Figure 3F** an increase in VE-Cadherin positive apoptotic cells in ventilated PDGF-R α haploinsufficient mice when compared to ventilated wildtype mice, confirming the finding in figure **3E**.

The authors provide no mechanism by which reduced PDGF signaling leads VEGF-A expression and endothelial apoptosis.

We undertook additional experiments in order to strengthen the mechanistic aspect of our manuscript. The activation of STATs through receptor tyrosine kinases like EGFR and PDGFR via recruitment of JAK has been outlined in the review ‘Hua Yu and Richard Jove; *THE STATS OF CANCER — NEW MOLECULAR TARGETS COME OF AGE*; *Nature Reviews cancer*; Volume 4; February 2004.’ Additionally, the direct activation of VEGF through activated STATs has been shown in ‘Guilian Niu *et al.*; *Constitutive Stat3 activity up-regulates VEGF expression and tumor angiogenesis*; *Oncogene* (2002) 21, 2000-2008.’ We therefore investigated the JAK/STAT pathway in our nCLD model. The results showed a downregulation of JAK-2 and STAT-3 together with decreased receptor expression in ventilated PDGF-R α haploinsufficient mice (**Figure 2H, I**) as well as in MFBs isolated from lungs of newborns (**Figure 6B, C**). Conversely, these proteins are upregulated upon PDGF-A treatment of PDGF-R α haploinsufficient prior to ventilation (**Figure 4E and F**). These data add to the link between decreased PDGF-R α expression and downregulation of VEGF-A via JAK-2/STAT-3.

In addition to reduced VEGF production involving the JAK/STAT pathway, we inferred a secreted factor produced by MFBs with attenuated PDGF signaling triggers endothelial cell apoptosis, using an siRNA and conditioned medium approach. We now show in **Figure 3J, K** a significant increase of cleaved caspase 9 and reduction in eNOS expression in endothelial cells upon treatment of these cells with the supernatant of PDGF-R α siRNA treated myofibroblasts. The siRNA treated myofibroblasts were characterized by reduced PDGF-R α and VEGF-A protein expression. The regulation of VEGF / eNOS / caspase 3 & 9 and its connection to endothelial cell death has been described by ‘Ningling Kang-Decker *et al.*; *Nitric oxide promotes endothelial cell survival Signaling through S-nitrosylation and activation of dynamin-2*; *Journal of Cell Science* 120; November 2006; 492-501’ and ‘Ziad Taimah *et al.*; *Vascular endothelial growth factor in heart failure*; *Nat. Rev. Cardiol.* 10; 519–530 (2013)’.

â€œ The author's postulate that administration of PDGF-A increases PDGF-R α in a feed forward mechanism. Is there any data to support this? This is important in order to validate use of PDGF-A as a therapy. It may be useful to assess global and cell specific gene expression in these experiments.

According to the review by 'Carl-Henrik Heldin; Targeting the PDGF signaling pathway in tumor Treatment; Cell Communication and Signaling 2013; 11:97,' and a study by 'Wang et al.; Platelet-derived Growth Factor Receptor-mediated Signal Transduction from Endosomes; Journal of biological chemistry; Vol. 279, No. 9; Issue of February 27, pp. 8038–8046, 2004', one of the mechanisms by which PDGF-R α is recycled through endosomes rather than being degraded in proteasomes or lysosomes is through activation and upregulation of signaling pathways like AKT. We therefore asked if provision of PDGF-A to haploinsufficient mice affect AKT. As shown in **Figure 4I**, AKT is upregulated, supporting an increase in PDGF-R α due to increased protein recycling and less degradation. In **Figure 7B** we now put forward the model showing activation of PDGF-R α through AKT upon PDGF-A treatment leading to activation of downstream pathways including JAK-2, STAT-3 and VEGF-A.

It is not clear why lung volumes were not impacted if pulmonary compliance improved with PDGF-A treatment, since the two should be related.

We agree with the reviewer, that in end-stage lung disease lung volume and compliance show a close correlation. In our case, the histologic analysis demonstrates an improvement in lung structure in PDGF-A treated mice. These structural changes are partially translated into a change in lung function, i.e. compliance. Nonetheless, the change in the peripheral lung architecture does not have to be mirrored by changes in lung volume occurring with more dramatic remodeling processes. In contrast, comparable lung volumes allow us to interpret histologic changes as a structural improvement rather than a change in the level of inflation.

Referee #2:

The authors investigated the role of PDGF-signaling in neonatal Chronic Lung Disease (nCLD) using different experimental system, such as samples from ventilated human newborns, samples from transgenic mice with PDGF-R α haplo-insufficiency (PDGF-R α ^{+/−}) and preclinical mouse model of mechanical ventilation with oxygen-rich gas (MV-O₂). Oak et al. suggest that PDGF-R α is a crucial molecular driver necessary for alveolar septation and vasculature integrity. In line with their model, exogenous PDGF-A rescues both structural and functional defects in the different experimental systems used for analysis of nCLD. Further, the authors show a decrease of PDGF-R α in some of the nCLD experimental systems, which correlates with an increase of TGF- β -signaling, thereby suggesting a counteracting causal involvement of both signaling pathways. In addition, the authors suggest a protective role of PDGF-R α for infants that develop nCLD. Since nCLD is the most common complication of preterm birth, the potential for clinical application of this work is compelling. Furthermore, even if a correlation between PDGF-signaling and nCLD has been previously reported (PMID: 24907056), the work presented by Oak et al. is conceptually novel because it explores in detail upstream- and downstream-effectors mediating the attenuation of PDGF-signaling in nCLD. Of special interest is the counteracting effect of PDGF- and TGF- β -signaling in the context of nCLD. Due to the potential clinical application and the conceptual novelty of the work, I would like to suggest this manuscript for publication at EMBO molecular medicine (EMM) after major revision. In the following lines, I will elaborate my concerns and suggestions to the authors not only to reach the standards of publication at EMM, but also to strengthen the experimental evidences supporting their model thereby increasing the impact of the work in the scientific community:

Major concerns:

1. The authors should include a supplementary table with the clinical relevant data of patients with nCLD analyzed in the manuscript. For instances, gender, gestational age (av.), weight (av.), country of maternal origin, ventilation and O₂ days (av.), severity of disease, survival, etc. This information will be useful for other scientists in the future implementing the results by Oak et al. in to their work.

The authors should write into the manuscript the number of approval by the ethic committee for the use of human samples for scientific purposes.

We thank reviewer for this helpful suggestion. We have now included a supplementary table (**Appendix Table. 3A-C**) reporting clinical variables of the patients included in the studies i.e. data in **Figure 1 D and E** and patients used for generating **Figure 5A** (negative correlation between PDGF-R α and TGF- β) and patients used to isolate fibroblasts from tracheal aspirates (**Figure 5I-M and 6L-M**). The patient characteristics for the GWAS analysis are already mentioned in the Appendix methods part under SNP analysis. We have also added the number of approval by the ethics committee.

2. The authors should deposit the array-based transcriptome analysis into a public accessible data base, for instances at the NCBI. During the review process, they should provide "reviewers-only" access in order that the quality of the data can be determined before publication.

Unfortunately, our informed consent does not allow us to provide this data in a publicly available data base. As the transcriptomic data only support the translation of the most important observations but were not used to generate a central hypothesis, we did not try to re-allocate the informed consents from the families at this point, although we do agree that the availability of the data should be routine in the future.

In addition, the results from the array-based transcriptome analysis should be confirmed by qRT-PCR based expression analysis of several of the target genes of the PDGF-signaling pathway. Perhaps some of the nominally associated genes presented in fourth column of the Suppl. Table 2 for the analysis of SNPs in PDGF-related pathways could be good options for the suggested qRT-PCR based expression analysis.

We appreciate the importance of validating the results of the gene expression microarrays. We did confirm the mRNA results at the protein level in a number of in vivo and in vitro experiments. However, due to stringent limits on sample volume obtained from preterm infants, we restricted our corroboration of the microarray results with staining of human lung tissue and experiments performed in human cells, and were not able to also carry out qRT-PCR confirmation.

3. The experimental evidences presented by Oak et al. supporting the causal involvement of enhanced TGF-B-signaling mediating the effects caused by reduced (or attenuated) PDGF-signaling during nCLD are relatively weak. Nevertheless, this is one of the attractive mechanistic aspects of the model proposed by the authors. The authors could improve this weak point performing the following experiments:

3.1 In Western Blot analysis (WB) of protein extracts after administration of PDGF-A in the Figure 4D, the authors should monitor the level of activity of TGF-B-signaling. For example using antibodies (Ab) specific against phosphorylated SMAD proteins and/or against proteins encoded by genes that are known targets of TGF-B-signaling. Based on the author's model, the administration of PDGF-A should reduce TGF-B-signaling.

Data now presented in the online supplement demonstrate similar levels of pSMAD2 in the lungs of ventilated wildtype as well as PDGFR α haploinsufficient mice, confirmed by quantitative fluorescence microscopy (**Expanded view Figure EV3A-B**). Underscoring the role of PDGFR α and its downstream signaling in neonatal lung injury upon ventilation, apoptosis in general (including MFBs) and in endothelial cells is increased in ventilated PDGFR α heterozygous mice versus wildtype littermates (**Expanded view Figure EV3C**). With the activation of TGF- β signaling only being a cause of decreased PDGFR signaling in ventilated mice and preterms, the treatment with the ligand PDGF-A does not require alteration of the TGF pathway for its effect. Instead we show an upregulation in PDGF-R α signaling (JAK2, STAT3) and expression upon PDGF-A treatment, as well as on VEGFA expression and endothelial cell survival (**Figures 2H, I, 4E, F and 6B, C**).

3.2 The inhibition of TGF-B-signaling in similar experimental settings as presented in the Figure 4 should have similar effects as the administration of PDGF-A, thereby supporting the model presented by the authors. There is several ways to inhibit the TGF-B-signaling. The author should select one or two ways to reduce (or block) TGF-B-signaling in the experimental settings presented in the Figure 4.

In the same line of ideas, the most elegant and striking demonstration of the causal involvement of TGF-B-signaling mediating the effects caused by attenuated PDGF-signaling during nCLD would be a rescue of the phenotype observed in the transgenic PDGF-R1 $\pm\pm$ mice by crossing this line with a second transgenic mice line, in which the TGF-B-signaling is attenuated.

However, I recognize that the generation of this double-transgenic mice line will require additional time and efforts that are out of the scope of the present manuscript.

We agree with the reviewer that additional inhibition of TGF- β is an appealing approach to prevent or ameliorate injury in the developing lung. Using gene-targeted mice to reduce TGF signaling for the purposes of our ventilation experiments may be technically challenging. As mentioned in the study by Sull M et. al., *Targeted disruption of the mouse transforming growth factor- β 1 gene results in multifocal inflammatory disease; Nature 359, 693 - 699 (22 October 1992)*, TGF- β null mice are systemically ill and die from wasting syndrome by 20 days of age. Combining this mutation with PDGF-R α haploinsufficiency may have additional developmental affects that could complicate the analysis. In any case, we are pursuing this experiment with conditional (rather than germline) deleted strains in the hopes we can eventually carry out this tantalizing genetic experiment. With respect to pharmacologic means of manipulating TGF- β in vivo, unfortunately the protein is too large for success by endotracheal delivery, and in any case would need to be inhibited. Some of our collaborators have already begun undertaking such approaches but have found them to be technically challenging, so they will require additional efforts to advance.

3.3 In addition to the immunostaining presented in the Figure 5A, the authors should monitor the activation of TGF-B-signaling by WB analysis of protein extracts using Ab specific against pSMAD2. The possibility of determining the MW during WB analysis will further support the specificity of the results obtained using this Ab. In addition, comparisons of the levels of proteins between different samples are more accurate by WB than by immunostaining.

While we agree this is a good experiment, we unfortunately do not have biopsy samples from the patients for Western blot analysis. In light of the severe limitation of human nCLD lung tissue samples, we instead performed pSMAD2 analysis on our precious few tissue sections from preterm infants with BPD. By immunostaining, we show a clear downregulation of PDGF-R α and a contrasting upregulation of pSMAD-2 in these patients. We also performed analysis on whole blood specimens, which corroborated these results (Figure 5A-B).

3.4 In the Figure 5D, the authors present a LUC-assay using a reporter-constructs containing part of the PDGF-R α promoter. First, the author should specify the region of the promoter cloned into the reporter plasmid. Second, a qRT-PCR based expression analysis of the endogenous PDGF-R α gene without or with TGF-B-treatment will be more relevant than the presented LUC-assay.

We have now indicated the SPAN of the promoter region that was cloned into the pGL4.14 plasmid in the methods part of the Appendix data submitted along with the manuscript previously. Additionally, we now demonstrate in Appendix Figure S5 the construct of the PDGF-R α promoter region used for cloning.

In addition, I strongly disagree with the description of the results obtained in this LUC-assay. The authors write: "...To test for a direct interaction between TGF- β and PDGF-R α , we conducted a luciferase assay by transfecting CCL206 cells with a pGal vector carrying a PDGF-R α promoter insert. Administration of TGF- β caused a 50% reduction in luciferase activity (Figure 5D), indicating that TGF- β directly inhibits PDGF-R α gene transcription, and supporting that the inverse relationship observed in nCLD is causal...".

A direct interaction between two proteins could be demonstrated by Co-immunoprecipitation assays, protein pull down experiments, proximity ligation assays, among others, but not with the LUC-assay presented by the authors. Further, the direct inhibition of a gene could be demonstrated by Chromatin immunoprecipitation (ChIP) assays, electrophoretic mobility shift assays (EMSA) in combination with expression analysis. I recommend the authors to describe this experiment in an appropriate manner.

We acknowledge the suggestion given by reviewer and have now modified the description accordingly to 'To test the impact of TGF- β on PDGF-R α promoter activity, we conducted a luciferase assay by transfecting CCL206 cells with a pGal vector carrying a PDGF-R α promoter insert. Administration of TGF- β caused a 50% reduction in luciferase activity (Figure 5D), indicating that TGF- β affects PDGF-R α gene transcription, supporting the inverse relationship observed in nCLD is causal. (p. 14). We apologize for the confusion in using the term "direct interaction," which we agree implies a physical interaction. Rather, we meant to indicate a cell-intrinsic, direct effect, and have clarified this in the manuscript.

4. In Figure 4E-F there is a dramatic increase of VE-Cadherin and VEGF-A upon PDGF-A treatment. The authors should evaluate the angiogenesis process in the lungs of ventilated mice, perhaps by endothelial cell tube formation assay or BrdU incorporation. The results will further support the model presented by the authors in the Figure 7, in which VEGF-A signaling is downstream of PDGF-signaling.

We appreciate the suggestion of the reviewer. However, because the normal process of microvascular maturation in the lung involves dynamic temporal and spatial heterogeneity, the analysis would likely be too complex. We therefore approached strengthening our model by fleshing out the mechanism by which PDGF signaling promotes VEGF signaling, and have included new data showing an increase in PDGF-R α downstream signaling, i.e. JAK/STAT and consequent upregulation of VEGF-A (Figure 2H-I, 4E-F, 6B-C). This observation is in line with our results presented for abrogated PDGF signaling in the injury model, where we successfully linked the lack of VEGF-A to endothelial cell apoptosis through an increase in caspase 3 and 9 signaling and reduced eNOS protein (Figure 3G,J-K). Indeed, the role of VEGF-A as a prosurvival factor for endothelial cells is well recognized.

The author should include into the model presented in the Figure 7 the proposed effect of TGF-B-signaling.

We have now revised the model in Figure 7 extensively showing the mechanistic link between PDGF-R α and VEGFA as well as effect of TGF-B through PDGF-R α on VEGFA and migration of MFBs primarily regulated by RAS/pERK signaling.

The author should describe the model presented in the Figure 7 at the beginning of the Discussion section. The last sentence in the current Results section could be used as introductory sentence for the description of the model in the Discussion.

We have now modified the discussion as suggested and, in addition, updated the model in Figure 7 to include the effect of TGF- β as well as the treatment effects.

5. By comparing the positions on the chromosomes of the SNPs in Figure 1A (n=117) with the position on the chromosomes of annotated SNPs from previously published data sets using the Genome Browser and the human genome hg19 (n=147), I detected not only differences in the numbers of SNPs (n=117 vs. n=147), but also in their median distribution (the majority of peaks is located nearer PDGF-RA 5' UTR). The authors should discuss the possible reasons for the observed discrepancies. In addition, the authors should explain why they used the human genome hg18 instead of hg19.

We have used hg18 data as the standard at the time of analysis. If required from the reviewers, we can lift positions of SNPs to hg38, the current genome build, or to hg19. We report a panel of 117 SNPs. For the region covered, more than 1,700 SNPs are reported according to the UCSC table browser (hg18). SNPs reported by us were genotyped on a custom genotyping array and passed the quality control criteria. Hence, any distributions do not necessarily have to match the results obtained exactly due to the array and quality criteria bias.

6. The colors for all the figures should be selected in order that "color-blind" individuals can still interpret the information. The combination of green and red in the immunostaining is not appropriate for color-blindness. They authors should refer to literature dealing with color combinations appropriate for color-blind individuals

The colors for the Figure 1 A should be selected in a similar manner and in order that the different groups can be recognized easier.

We appreciate this concern of the reviewer. Due to the presentation of more than two colors in the immunostaining figures, we are left with no appropriate option to demonstrate the difference clearly enough. Therefore, we now provide greyscale images in **related file for referees 1** with arrows to indicate the differences and have added distinct arrows in the figure from the main manuscript.

7. The authors should use scale bars in all microscopy pictures (including the immunostainings) and describe them in the legend.

We have added a representative immunofluorescence image that includes a scale bar in the anuscrypt.

8. The author should increase the size of the pictures from the WB presented in all the figures starting from Figure 2 onwards. If there are complications due to limited space in the figures, the authors could consider presenting several WB pictures together in the same panel. For example, the WB pictures presented in the panels 2B and 3B-D could be presented together, since they are from the same experiment using different Abs during the immunodetection after WB. In a similar manner, the WB in the panels 3H and 3J could be presented together. The WB in the panels 4D-F could be presented together. The WB in the panels 5E-G could be presented together.

We now show in **related file for referees 2**, enlarged western blot images used throughout the entire manuscript.

If the results are robust, I rather prefer to see the WB pictures than the densitometry analysis. In addition, the pictures of the WB selected for the figure should correlate with the densitometry analysis. This is not always the case in the current manuscript. For example, in the Figure 2G the mean of the WT group in the plot is above 1.0, what would mean that the intensity of ACTB bands is lower than the intensity of the PDGF-R α bands. The opposite is the case in the pictures used for the Figure 2G.

We thank reviewer for pointing this out. In order to address the concern of the reviewer we now provide **related file for referees 2** with enlarged western blots. We have also checked and revised the manuscript for appropriate representative western blot images and quantification including **Figure 2G**.

9. The authors have to improve the description of the statistical analysis performed. The way how is described in the legends and in the Material and Methods section of the current version of the manuscript does not allow the reader to recognize which analysis was performed in each panel to determine the statistical significance of the values presented. In addition, the authors should specify the method used to determine the distribution of the data set.

We apologize for the confusion. As requested, we have modified all the figure legends to indicate the statistical test that was used to determine significance.

Minor concerns:

10. The author should specify the company, catalog number of the Abs. used for immunostaining. The author should also specify the same information for the PDGF-A protein used in the manuscript.

We thank reviewer for pointing out the missing antibody catalog numbers. We have now revised and inserted catalog numbers for all the antibodies in appropriate sections.

11. The authors should consistently use in the manuscript the official names and official symbols for the proteins and the genes described, which can be found at NCBI. In addition, the authors should use the nomenclature standards for the description of proteins and genes in human and mouse. The research community has adopted and nomenclature standards for each model organism and published them on the relevant model organism websites and in scientific journals.

We appreciate the suggestion provided by the author. In order to prevent confusion with similar protein nomenclature, e.g. PDGF-RA (NCBI nomenclature) and PDGFA (NCBI nomenclature) we decided to use distinct terminology, i.e. PDGF-R α and PDGF-A. We now provide a table referring to the NCBI nomenclature (**related file for referees 3**).

12. When referring to proteins and specially to post-translational modified proteins as phosphorylated SMAD or ERK proteins, I strongly suggest the authors to substitute the word "expression" by "levels" or "protein levels".

We apologize for the confusion and have now changed the word 'expression' to 'protein level' or 'level' wherever appropriate.

13. There are several inconsistencies in the legends of the figures. For example in Figure 1 the legends of panels A and B are swapped. In Supp. Figure 1 there is not Panel C, as it is referred in Methods "Gene-targeted mice".

We thank reviewer for pointing out these inconsistencies, which we have corrected.

14. There are some typos in the text. I suggest the authors to run once again the spelling and grammar command from the text program used to detect them.

"Messages" should be substituted by "messengers" or "mRNAs", for instances in the Introduction, page 7.

"mics" should be substitute by "mice" in the legend of the Figure 2C.

The word "interaction" should be substituted by "cross-talk" when referring to signaling pathways.

Consider rephrasing the titles in the Result section. For instances, the current title "Enrichment of PDGFR-Î± SNPs and reduced expression of PDGF-RÎ± by human lung fibroblasts in ventilated preterm infants developing nCLD" could be substituted by "Enrichment of PDGFR-Î± SNPs and reduced expression of PDGF-RÎ± IN human lung fibroblasts FROM ventilated preterm infants developing nCLD"

We apologize for missing these typos and inappropriate word choices and have corrected them.

Referee #3:

This is an interesting packet of information looking at PDGF in alveolar simplification from several angles. First they show that there are some SNPs in the PDGF pathway that may be relevant.

Then they look at the response of PDGF pathway haplotypes to oxygen and ventilation.

Then they look at the intercurrent role of TGF-β pathway signaling in isolated myofibroblasts.

Lastly they show a rescue study with PDGF ligand.

This is a nice overall package although the way it is laid out is more confusing than it seems from what I have written here. So the manuscript could do with shortening and omg elf the data taking out as there are really several separate stories here I think.

We apologize for the confusion and have tried to improve the clarity and shorten the length in our revised version.

The other comment is that it is not clear to me what relevant e these studies may have to human other than the SNPs because it is becoming increasingly clear that mouse models of lung development and injury repair, although they are very interesting and convenient may not actually have a lot to do with the fine details of human development and repair.

We agree this is a significant point and therefore conducted extensive functional analysis of the SNPs on gene expression and MFB function. We are pleased to report that these experiments have confirmed the human relevance of our mouse studies, implicating attenuated PDGF signaling as a central driver of nCLD (**Figure 1C-G, Appendix Figure S1**).

I also think that PDG ligand rescue in human would be very tricky. Mouse studies of excess PDGF ligand are far from reassuring that this will result in normal lung development in mice so that getting the dose response and administration regimens right will be hugely challenging.

We completely agree with these concerns and that there is significant work to be done before embarking on clinical development. However, we are excited to have implicated attenuated PDGF-Rα signaling as a central driver of nCLD pathology, since now translational efforts can focus on devising ways to safely augment PDGF signaling which could be tremendously beneficial for screening for at risk preterm infants, of for preventing or treating nCLD.

The results are quite consistent with their hypothesis in the mouse system.

The challenge with this i that it is becoming increasingly clear that the mouse and human systems are not exactly the same. Thus the relevance of the very detailed findings in this paper are open to some question. This is a fairly recent development however. But I think it should at least be discussed.

As we describe above, we agree the importance of human relevance cannot be underestimated, so we are excited that our human SNP functional analysis confirmed a reduction in PDGF-Rα expression and that lung fibroblasts isolated from infants carrying the SNPs recapitulated the findings of our mouse experiments. We therefore feel the manuscript provides a very integrated

view of nCLD that fairly includes mechanistic insights drawn from our PDGF-R α haploinsufficient mouse model.

2nd Editorial Decision

12 April 2017

Thank you for the submission of your revised manuscript to EMBO Molecular Medicine. We have now received the enclosed reports from the reviewers that were asked to re-assess it. As you will see, while they are now globally supportive, reviewers 1 and 2 still have a number of concerns that require your action.

As you will see, many are (important) requests for clarification and appropriate toning down of some overstated claims. Among the concerns however, there are two especially important points: 1) to provide a much better description of the human samples and how they were obtained and 2) to make raw data available in a publicly-accessible databases.

Regarding the latter point, I'm afraid that we cannot accept your contention that this is not possible, as the reviewer also points out. In fact, for example, the European Genome-phenome Archive allows access control of datasets when needed. It is possible to submit information to the EGA while still continuing to manage access via a Data Access Committee (DAC): <https://www.ebi.ac.uk/ega/home>. It's important to stress that the DAC - which one would need to allow access to the raw data in some way - would remain unchanged. Many studies, each with managed access, do this (see: <https://www.ebi.ac.uk/ega/datasets>). I should also mention that it often takes quite a bit of time for submission. This can be fast tracked but it is around 3 or 4 weeks. This is because one's DAC needs to be set up, documentation submitted, etc. I quote from the EGA site:

"Who controls access to this dataset?"

For each dataset that requires access control, there is a corresponding Data Access Committee (DAC) who determines access permissions. Data access requests are reviewed by the relevant DAC, not by the EGA"

Eventually, the text in the manuscript would be something like this: "Our datasets were obtained from subjects who have consented to the use of their individual genetic data for biomedical research, but not for unlimited public data release. Therefore, we submitted it to the European Genome-phenome Archive, through which researchers can apply for access of the raw data." Please make sure you also update the Author Checklist accordingly.

Provided you address all remaining concerns fully and carefully, I am willing to make an editorial decision on your next, final version.

Please also apply the following final amendments:

- 1) Many of the figures are of insufficient size. Please provide figures of the appropriate size. You might consider referring to page 5 of our figure guide in our guidelines for authors (http://embopress.org/sites/default/files/EMBOPress_Figure_Guidelines_061115.pdf)
- 2) Please provide a scale bar for Fig. S3
- 3) We are still missing the ORCID number for Dr. Desai
- 4) The manuscript must include a statement in the Materials and Methods identifying the institutional and/or licensing committee approving the experiments, including any relevant details (like how many animals were used, of which gender, at what age, which strains, if genetically modified, on which background, housing details, etc). We encourage authors to follow the ARRIVE guidelines for reporting studies involving animals. Please see the EQUATOR website for details: <http://www.equator-network.org/reporting-guidelines/improving-bioscience-research-reporting-the-arrive-guidelines-for-reporting-animal-research/>. Please make sure that ALL the above details are reported, including in the Author checklist
- 5) We encourage the publication of source data, with the aim of making primary data more accessible and transparent to the reader. Would you be willing to provide a PDF file per figure that contains the original, uncropped and unprocessed scans of all or at least the key gels used in the

manuscript and/or source data sets for relevant graphs? The files should be labeled with the appropriate figure/panel number, and in the case of gels, should have molecular weight markers; further annotation may be useful but is not essential. The files will be published online with the article as supplementary "Source Data" files. If you have any questions regarding this just contact me.

6) Every published paper includes a 'Synopsis' to further enhance discoverability. Synopses are displayed on the journal webpage and are freely accessible to all readers. They include a short description as well as 2-5 one-sentence bullet points that summarise the key NEW findings of the paper. The bullet points should be designed to be complementary to the abstract - i.e. not repeat the same text. We encourage inclusion of key acronyms and quantitative information. Please use the passive voice. Please attach this information in a separate file or send them by email, we will incorporate it accordingly.

I look forward to reading a new revised version of your manuscript as soon as possible.

***** Reviewer's comments *****

Referee #1 (Comments on Novelty/Model System):

Use of a mouse in which the experiments are carried out on a cell type restricted PDGF-R α haplo-insufficient animal would make a more compelling argument for the myofibroblast as the target cell.

Referee #1 (Remarks):

The authors' have been for the most part responsive to my critiques that were submitted for the original version of the manuscript. Specifically they have provided additional human data linking PDGF-R α SNP to attenuated gene expression. In addition they have provided mechanistic detail linking PDGF signaling to alveologenesis and endothelial dysfunction. There remain however, a few minor issues, which require clarification.

- The authors should consider including a statement in the results that the data linking PDGF-R α SNPs to decreased gene expression was from cord blood and not from alveolar myofibroblasts.
- o Figure 1C needs to be amended. It does not indicate units in either the figure or the legend, one of the panels is mislabeled and all the panels with different minor alleles look identical.
- Their explanation on why their data suggests that the effects are myofibroblast mediated is not convincing. They should consider adding a statement that the data cannot exclude possibility that the effects may be independent of alveolar myofibroblasts (i.e. mediated by alveolar epithelial, endothelial or macrophage), because the haplosufficient mice are not cell specific deleted.
- The explanation for why lung volumes are unchanged (no statistics are offered) in the VEGF-A treatment experiment is not well explained. They should consider adding a statement to clarify this. Also need statistics for the volume measurement.
- They need to clarify for the experiments in the human samples were used, whether the samples were collected randomly, from consecutive patients or a convenience sample.

Referee #2 (Remarks):

The authors discussed most of the concerns (not only my concerns, but also the ones of the other Reviewers) using arguments based on the literature and/or interpretation on their data in the present manuscript. The authors also presented new human SNP analysis from a new cohort of preterm infants harboring the SNP, which not only corroborate the human relevance of their mouse studies, but also strongly support the causal involvement of attenuated PDGF signaling in the pathogenesis of nCLD. Overall I recognize the efforts of the authors that resulted in an improved manuscript that confirms my original positive opinion about the conceptual novelty and clinical relevance of the work.

It is a pity that the authors were not able to perform several experiments arguing stringent limits on sample volume obtained from preterm infants. Even though I can understand this limitation, there is

a critical aspect that I would like to emphasize: the raw data of the transcriptome analysis should be available first for quality assessment of the data during the review process and later on for other scientists to increase the impact of the work in the field. Based on the answer of the authors to the second point of my previous Review, I would like to suggest the following options:

- Place the raw data in a public available data base anonymizing the donors. This is possible with most of the patients consents and the support of the local ethic commission.
- If the previous option is not possible, since the authors claim: "...As the transcriptomic data only support the translation of the most important observations but were not used to generate a central hypothesis...", the authors might consider to remove the transcriptome data from the manuscript and re-structure it accordingly.

I hope that the authors select the first option in order that the overall quality of the manuscript is not reduced. In addition, the confirmation of "omic-data" by single gene analysis should be a routine in our days, especially if the quality of the raw data is not optimal. However, due to the limitation on sample volume obtained from preterm infants one could obviate these experiments if the data show a minimum of quality. I hope that the authors and the editors understand my concern.

Referee #3 (Remarks):

This is now suitably revised. It's a novel set of ideas about PDGF signaling and BPD.

2nd Revision - authors' response

25 July 2017

Response to the Editor

General editorial comment: Requests for clarification and appropriate toning down of some overstated claims.

Answer: We appreciate the opportunity to revise some remaining statements that raised the concern of the reviewers. We have clarified remaining contents especially with respect to the patient cohorts (see below). Moreover, we have added in a discussion of the limitations regarding the interpretation of the data in order not to overstate our claims.

Among the concerns however, there are two especially important points:

#1) Provide a much better description of the human samples and how they were obtained.

Answer: We have significantly revised the methods section to provide a detailed description of the patients that provided the samples used for the different analyses. We now include specifics of the time points and the patient characteristics for the different groups (see p19-21, supplemental materials and methods, appendix methods, table S3).

#2) Make raw data available in a publicly-accessible databases.

*Answer: We have obtained the approval of the respective ethics committees and were now given access to upload the original transcriptome data to the European Genome-Phenome Archive (EGA) under accession number **EGAS00001002586**. Dr. Windhorst (Anita.C.Windhorst@informatik.med.uni-giessen.de) is in the process of uploading the data. Please do not hesitate to contact her or us for further information such as details of the submission for your information. Additionally we have now added the following sentence to our manuscript: "Our datasets were obtained from subjects who have consented to the use of their individual genetic data for biomedical research, but not for unlimited public data release. Therefore, we submitted it to the European Genome-phenome Archive (study unique name-ena-STUDYIMI-24-07-2017-10:03:30:362-576), through which researchers can apply for access of the raw data." (see p20). We have also updated the Author Checklist accordingly.*

#1) Many of the figures are of insufficient size. Please provide figures of the appropriate size. You might consider referring to page 5 of our figure guide in our guidelines for authors

http://embopress.org/sites/default/files/EMBOPress_Figure_Guidelines_061115.pdf)

Answer: We apologize for the inconvenience caused by low resolution of the figures and have now revised all the figures to improve clarity and enlarge size. The new improved figures are uploaded for publication.

#2) Please provide a scale bar for Fig. S3

Answer: As suggested by the editor, we incorporated a scale bar in figure S3.

#3) We are still missing the ORCID number for Dr. Desai

Answer: Dr. Desai has registered with ORCID and associated his number with this manuscript.

#4) The manuscript must include a statement in the Materials and Methods identifying the institutional and/or licensing committee approving the experiments, including any relevant details (like how many animals were used, of which gender, at what age, which strains, if genetically modified, on which background, housing details, etc). We encourage authors to follow the ARRIVE guidelines for reporting studies involving animals. Please see the EQUATOR website for details: <http://www.equatornetwork.org/reporting-guidelines/improving-bioscience-research-reporting-the-arrive-guidelines-forreporting-animal-research/>. Please make sure that ALL the above details are reported, including in the Author checklist

Answer: According to the ARRIVE guidelines for mouse handling and experimentation, we have now added statements regarding randomization and laboratory environment in the methods part of the manuscript (see p24). The further description is in accordance with the guidelines explaining allocation of the animals per group as well as maintenance of the mouse line.

#5) We encourage the publication of source data, with the aim of making primary data more accessible and transparent to the reader. Would you be willing to provide a PDF file per figure that contains the original, uncropped and unprocessed scans of all or at least the key gels used in the manuscript and/or source data sets for relevant graphs? The files should be labeled with the appropriate figure/panel number, and in the case of gels, should have molecular weight markers; further annotation may be useful but is not essential. The files will be published online with the article as supplementary "Source Data" files. If you have any questions regarding this just contact me.

Answer: We are herewith submitting the PDF file containing uncut blots to be made accessible to the reader for more transparency.

#6) Every published paper includes a 'Synopsis' to further enhance discoverability. Synopses are displayed on the journal webpage and are freely accessible to all readers. They include a short description as well as 2-5 one-sentence bullet points that summarise the key NEW findings of the paper. The bullet points should be designed to be complementary to the abstract -i.e. not repeat the same text. We encourage inclusion of key acronyms and quantitative information. Please use the passive voice. Please attach this information in a separate file or send them by email, we will incorporate it accordingly.

Answer: We herewith submit a separate synopsis file.

Response to the Reviewer

Referee #1 (Comments on Novelty/Model System):

Use of a an mouse in which the experiments are carried out on a cell type restricted PDGF-R α haploinsufficient animal would make a more compelling argument for the myofibroblast as the target cell.

Answer: We agree that conditional deletion is always preferable to germline deletion for strengthening the proof of cell type specificity, but a flox'd PDGFR α mouse strain is not available to our knowledge. Also, there is an abundance of previous evidence implicating PDGF signaling in myofibroblasts as essential for alveolarization at the stages we are studying

(from analysis of various PDGF receptor and ligand knockout strains), and the myofibroblast is known to be the primary cell type expressing PDGFR α in the gas exchange region at the time of our analysis. Given this prior knowledge, and since our ex vivo studies on isolated myofibroblasts reproduce the results predicted from our whole animal experiments in vivo, we do not think it is essential for our conclusions to generate a new mouse strain. Our studies on human fibroblasts are also fully concordant, providing independent support for our inference.

Referee #1 (Remarks):

The authors' have been for the most part responsive to my critiques that were submitted for the original version of the manuscript. Specifically they have provided additional human data linking PDGFR α SNP to attenuated gene expression. In addition they have provided mechanistic detail linking PDGF signaling to alveologenesis and endothelial dysfunction. There remain however, a few minor issues, which require clarification.

#1 The authors should consider including a statement in the results that the data linking PDGF-R α SNPs to decreased gene expression was from cord blood and not from alveolar myofibroblasts.

Answer: We thank reviewer for catching this oversight and have corrected the text to indicate specifically studies involving human blood versus myofibroblasts (see p8).

#2 Figure 1C needs to be amended. It does not indicate units in either the figure or the legend, one of the panels is mislabeled and all the panels with different minor alleles look identical.

Answer: We thank reviewer for pointing out the missing units. In the revised figure we have labelled the units (log₂ gene expression), corrected the mis-spelled labels (homozygous instead of homozygote) and have amended the figure to show the data the overlapping data as well. All panels with different minor alleles look identical because if one patient shows a minor allele in one SNP, then it also has at least one minor allele in the other SNPs.

#3 Their explanation on why their data suggests that the effects are myofibroblast mediated is not convincing. They should consider adding a statement that the data cannot exclude possibility that the effects may be independent of alveolar myofibroblasts (i.e. mediated by alveolar epithelial, endothelial or macrophage), because the haploinsufficient mice are not cell specific deleted.

Answer: The reason we specifically implicate the myofibroblast is that it is the only alveolar cell type that expresses PDGFR α at the experimental time points, and indeed no other alveolar lineage (including epithelial, endothelial, and macrophage) expresses PDGFR α at any time during development. Therefore, since the knockout only impacts PDGFR α , it is formally impossible that any of these other alveolar cell types is responsible for the phenotype. This specificity is the very reason we used a genetic model for our studies, because it allows us to confidently ascribe the phenotype to the absence of PDGFR α , since no other gene in the entire genome is altered. However, we do appreciate that PDGFR α is developmentally expressed by other cells in and outside of the lung, and that we did not delete it specifically in myofibroblasts at the time of the experimental manipulation. Thus, while we can confidently implicate PDGFR α -expressing cells and exclude other cell lineages as the underlying cause of the phenotype, it is possible this could involve effects on cells that expressed PDGFR α earlier in development. We have added this caveat to our discussion.

#4 The explanation for why lung volumes are unchanged (no statistics are offered) in the VEGF-A treatment experiment is not well explained. They should consider adding a statement to clarify this. Also need statistics for the volume measurement.

Answer: We apologize for this oversight and have now included statistical analysis showing unchanged lung volumes in the groups undergoing PDGF-A treatment (see p12). In order to interpret the presence of enlarged and fewer alveoli as a consequence of lung apoptosis in contrast to the reason being simple overinflation due to altered lung compliance, data for lung volume need to be taken into account. Unchanged lung volumes together with the given alterations in lung structure indicate alveolar simplification resulting from lung apoptosis. This is now added in the revised version of the manuscript.

#5 They need to clarify for the experiments in the human samples were used, whether the samples were collected randomly, from consecutive patients or a convenience sample.

Answer: We thank reviewer for pointing out the missing information. We added a sentence in the methods part (see p20) stating that the samples were randomly collected from patients available for analysis. With respect to the protein analysis, patients for the non-BPD group were chosen from the patients available in order to match for gestational age, birth weight, gender and initial respiratory disease.

Referee #2 (Remarks): EMM-2016-07308 Oak P, Pritzke T et. al., 2016 , 2nd Review

The authors discussed most of the concerns (not only my concerns, but also the ones of the other Reviewers) using arguments based on the literature and/or interpretation on their data in the present manuscript. The authors also presented new human SNP analysis from a new cohort of preterm infants harboring the SNP, which not only corroborate the human relevance of their mouse studies, but also strongly support the causal involvement of attenuated PDGF signaling in the pathogenesis of nCLD. Overall I recognize the efforts of the authors that resulted in an improved manuscript that confirms my original positive opinion about the conceptual novelty and clinical relevance of the work.

It is a pity that the authors were not able to perform several experiments arguing stringent limits on sample volume obtained from preterm infants. Even though I can understand this limitation, there is a critical aspect that I would like to emphasize: the raw data of the transcriptome analysis should be available first for quality assessment of the data during the review process and later on for other scientists to increase the impact of the work in the field. Based on the answer of the authors to the second point of my previous Review, I would like to suggest the following options:

#1 -Place the raw data in a public available data base anonymizing the donors. This is possible with most of the patients consents and the support of the local ethic commission.

-If the previous option is not possible, since the authors claim: "...As the transcriptomic data only support the translation of the most important observations but were not used to generate a central hypothesis...", the authors might consider to remove the transcriptome data from the manuscript and re-structure it accordingly.

I hope that the authors select the first option in order that the overall quality of the manuscript is not reduced. In addition, the confirmation of "omic-data" by single gene analysis should be a routine in our days, especially if the quality of the raw data is not optimal. However, due to the limitation on sample volume obtained from preterm infants one could obviate these experiments if the data show a minimum of quality. I hope that the authors and the editors understand my concern.

Answer: We have now acquired the permission from respective ethical committees and are in process of uploading the raw data in EGA public library (accession number-EGAS00001002586). Our colleague Dr. Windhorst is uploading the data and is available on Anita.C.Windhorst@informatik.med.uni-giessen.de in case of further questions regarding the progress. "Our datasets were obtained from subjects who have consented to the use of their individual genetic data for biomedical research, but not for unlimited public data release. Therefore, we submitted it to the European Genome-phenome Archive (study unique nameena-STUDY-IMI-24-07-2017-10:03:30:362-576), through which researchers can apply for access of the raw data." (see p20). We have also updated the Author Checklist accordingly.

Referee #3 (Remarks):

This is now suitably revised. It's a novel set of ideas about PDGF signaling and BPD.

Corresponding Author Name: Dr. Anne Hilgenorff

Manuscript Number: EMM-2016-07308